# Unravelling the operation of organic artificial neurons for neuromorphic bioelectronics

Pietro Belleri [1], Judith Pons i Tarrés [2], Iain McCulloch [3], Paul W. M. Blom [2], Zsolt M. Kovács-Vajna [1], Paschalis Gkoupidenis [2,4,5] ✉ & Fabrizio Torricelli [1] ✉

Organic artificial neurons operating in liquid environments are crucial components in neuromorphic bioelectronics. However, the current understanding of these neurons is limited, hindering their rational design and development for realistic neuronal emulation in biological settings. Here we combine experiments, numerical non-linear simulations, and analytical tools to unravel the operation of organic artificial neurons. This comprehensive approach elucidates a broad spectrum of biorealistic behaviors, including firing properties, excitability, wetware operation, and biohybrid integration. The non-linear simulations are grounded in a physics-based framework, accounting for ion type and ion concentration in the electrolytic medium, organic mixed ionic-electronic parameters, and biomembrane features. The derived analytical expressions link the neurons spiking features with material and physical parameters, bridging closer the domains of artificial neurons and neuroscience. This work provides streamlined and transferable guidelines for the design, development, engineering, and optimization of organic artificial neurons, advancing next generation neuronal networks, neuromorphic electronics, and bioelectronics.

Neuromorphic electronics aim at the realization of intelligent systems that emulate the immense capability of the nervous system to efficiently cope with a wide diversity of environmental and biological signals[1–3]. By emulating the brain's basic building blocks such as synapses and neurons, neuromorphic electronics can perform bio-inspired processing and computation, opening opportunities in a broad range of application fields, including edge computing[4], wearables[5], point-of-care diagnostics[6,7], bioelectronics[8,9], (bio) robotics and environmental intelligence[10]. A wide range of materials has been used over the past decade for the realization of neuromorphic electronics including metal oxides[11–16], ferroelectrics[17,18], ferromagnetics[19,20], phase change[21–23], and 2D materials[24,25].

Neuromorphic electronics made of organic materials are of particular interest because of their close resemblance with biology[26].

Neuromorphic electronics with soft electrochemical matter such as organic mixed ionic electronic conductors (OMIECs), can emulate realistically biological phenomena because of their responsiveness to the biological carriers of information (alkaline ions, neurotransmitters, neuromodulators, etc.)[26–30]. Moreover, their operation in wet biological environments opens entirely new directions for neuromorphic biosensors and bioelectronics. Indeed, over the past few years, the basic building blocks of the nervous system have been realized with OMIECs and their corresponding devices (i.e., organic electrochemical transistors or OECTs) such as low-voltage artificial synapses and synaptic networks[31,32].

Organic electrochemical artificial neurons (OANs) are the latest entry of building blocks, with a few different approaches for circuit realization. OANs possess the remarkable capability to realistically

[1]Department of Information Engineering, University of Brescia, via Branze 38, 25123 Brescia, Italy. [2]Max Planck Institute for Polymer Research, Ackermannweg 10, 55128 Mainz, Germany. [3]Department of Chemistry, University of Oxford, 12 Mansfield Road, Oxford, UK. [4]Department of Electrical and Computer Engineering, North Carolina State University, 890 Oval Dr, Raleigh, NC, USA. [5]Department of Physics, North Carolina State University, 2401 Stinson Dr, Raleigh, NC, USA. ✉e-mail: gkoupidenis@mpip-mainz.mpg.de; fabrizio.torricelli@unibs.it

mimic biological phenomena by responding to key biological information carriers, including alkaline ions, noise in the electrolyte, and biological conditions. An organic artificial neuron with a cascade-like topology made of OECT inverters has shown basic (regular) firing behavior and firing frequency that is responsive to the concentration of ionic species ($Na^+$, $K^+$) of the host liquid electrolyte[33]. An organic artificial neuron consisting of a non-linear building block that displays S-shape negative differential resistance (S-NDR) has also been recently demonstrated[34]. Due to the realization of the non-linear circuit theory with OECTs and the sharp threshold for oscillations, this artificial neuron displays biorealistic firing properties and neuronal excitability that can be found in the biological domain such as input voltage-induced regular and irregular firing, ion and neurotransmitter-induced excitability and ion-specific oscillations. Biohybrid devices comprising artificial neurons and biological membranes have also shown to operate synergistically, with membrane impedance state modulating the firing properties of the biohybrid in situ. More recently, a circuit leveraging the non-linear properties of antiambipolar OMIECs, which exhibit negative differential transconductance, has been realized[35]. These neurons show biorealistic properties such as various firing modes and responsivity to biologically relevant ions and neurotransmitters. With this neuron, ex-situ electrical stimulation has been shown in a living biological model. Therefore, the class of OANs perfectly complements the broad range of features already demonstrated by solid-state spiking circuits (Supplementary Table 1), offering opportunities for both hybrid interfacing between these technologies and new developments in neuromorphic bioelectronics.

Despite the promising recent realizations of organic artificial neurons, all approaches still remain in the qualitative demonstration domain and a rigorous investigation of circuit operation is still missing. Indeed, quantitative models exist only for inorganic, solid-state artificial neurons without the inclusion of physical soft-matter parameters and the biological wetware (i.e., aqueous electrolytes, alkaline ions, biomembranes)[36,37]. This gap in knowledge significantly impedes the simulation of larger-scale functional circuits, and therefore the design and development of integrated organic neuromorphic electronics, biohybrids, OAN-based neural networks, and intelligent bioelectronics.

In this work, we unravel the operation of organic artificial neurons that display non-linear phenomena such as S-shape negative differential resistance (S-NDR). By combining experiments, numerical simulations of non-linear iontronic circuits, and newly developed analytical expressions, we investigate, reproduce, rationalize, and design the wide biorealistic repertoire of organic electrochemical artificial neurons including their firing properties, neuronal excitability, wetware operation, and biohybrid formation. The OAN operation is efficiently rationalized to include how neuronal dynamics are probed by biochemical stimuli in the electrolyte medium. The OAN behavior is also extended on the biohybrid formation, with a solid rationale of the in situ interaction of OANs with biomembranes. Non-linear simulations of OANs are rooted in a physics-based framework, considering ion type, ion concentration, organic mixed ionic–electronic parameters, and biomembrane properties. The derived analytical expressions establish a direct link between OAN spiking features and its physical parameters and therefore provide a mapping between neuronal behavior and materials/device parameters. The proposed approach open opportunities for the design and engineering of advanced biorealistic OAN systems, establishing essential knowledge and tools for the development of neuromorphic bioelectronics, in-liquid neural networks, biohybrids, and biorobotics.

## Results

The OAN is obtained by connecting an organic electrochemical non-linear device (OEND) with a biasing network comprising a resistor $R_L$, a capacitor $C_L$, and a DC voltage generator $V_{IN}$ (Fig. 1a). The OEND comprises two organic electrochemical transistors (OECTs), named $T_1$ and $T_2$, and two resistors, named $R_1$ and $R_2$. We used the mixed ionic–electronic conductors poly(3,4-ethylenedioxythiophene) doped with poly(styrene sulfonate) (PEDOT:PSS) and poly(2-(3,3-bis(2-(2-(2-methoxyethoxy)ethoxy)ethoxy)-[2,2-bithiophen]−5-yl) thieno [3,2-b] thiophene) (p(g2T-TT)) as channel materials for $T_1$ and $T_2$, respectively. $T_1$ is a normally on, viz. depletion-mode, OECT while $T_2$ is a normally off, viz. accumulation mode, OECT. Typical OECT transfer characteristics ($I_D - V_G$) measured at various drain voltages ($V_D$) are displayed in Supplementary Fig. 1 of the Supplementary Information.

To investigate the OAN, we model the OECT electrical characteristics (see Supplementary Note 1) accounting for material and device parameters such as the volumetric capacitance $C_V$[38,39], ion-concentration-dependent threshold voltage $V_{TH}$[40], energy disorder $\gamma$[41,42], and channel-length modulation[43,44]. Then, we implemented the OECT model in a circuit simulator. As displayed in Fig. 1b, c, we accurately reproduced the measured electrical characteristics of the OEND. Voltage vs. current operation mode of the OEND is a typical measurement conducted for characterizing non-linear devices, such as redox-diffusive memristors, Mott memristors, single-transistor latch, and Gaussian heterojunction transistors[45]. Serving as the non-linear core component of the OAN, the OEND exhibits a behavior similar to other spiking neuron devices based on non-linear elements, displaying an abrupt increase or decrease in either the current-voltage or voltage-current relationship. The current-voltage characteristic of the OEND is crucial for designing and shaping the spiking behavior of the OAN. Specifically, the OEND can be assessed either as $I_{OEND}(V_{OEND})$ where $V_{OEND}$ is the independent (input) variable (Fig. 1b), or as $V_{OEND}(I_{OEND})$ where $I_{OEND}$ is the independent variable (Fig. 1c). When the OEND is operated in voltage mode, $V_{OEND}$ is the independent input variable: $V_{OEND}$ is swept forward and backward and the current $I_{OEND}$ flowing through the OEND is recorded. As displayed in Fig. 1b, a hysteretic characteristic is obtained under steady-state operation, which is inherently attributed to the non-linear circuit configuration. Conversely, when the OEND is operated in current mode, $I_{OEND}$ is the independent input variable: $I_{OEND}$ is swept forward and backward and the voltage $V_{OEND}$ is recorded. As displayed in Fig. 1c the $V_{OEND}(I_{OEND})$ characteristic is non-hysteretic and shows S-shape negative differential resistance (S-NDR). A comprehensive discussion on OEND operation is provided later in the manuscript ("Organic electrochemical non-linear device operation" section).

Finally, we connected the OEND to the biasing network and we performed non-linear transient simulations of the OAN. As displayed in Fig. 1d, e the simulations predict both the experimental current and voltage output oscillations. The material and device parameters are listed in the "Methods" section unless otherwise stated. As we recently reported[34], the artificial action potentials of the OAN and its firing properties, including excitability, spike latency, oscillation frequency, and amplitude, are sensitive to the local electro-bio-chemical signals in the liquid environment, viz. the extracellular space.

## Organic electrochemical non-linear device operation

As a first step, we focus on the OEND, which is the core component of the OAN. When operating the OEND in voltage mode, $V_{OEND}$ serves as the independent input variable. $V_{OEND}$ is swept forward and backward, while recording the current $I_{OEND}$ flowing through the OEND. As illustrated in Fig. 2a, under steady-state conditions, a hysteretic $I_{OEND}(V_{OEND})$ characteristic is observed. This hysteresis is inherently due to the non-linear switching of the OETCs. The OEND internal voltages controlling the operation of the OECTs $T_1$ and $T_2$ are marked in Fig. 2b and are named $V_{GS1}$, $V_{SG2}$, and $V_{GD2}$. The OEND internal voltages calculated by means of numerical simulations are displayed in Fig. 2c, d and Supplementary Fig. 2. For the sake of

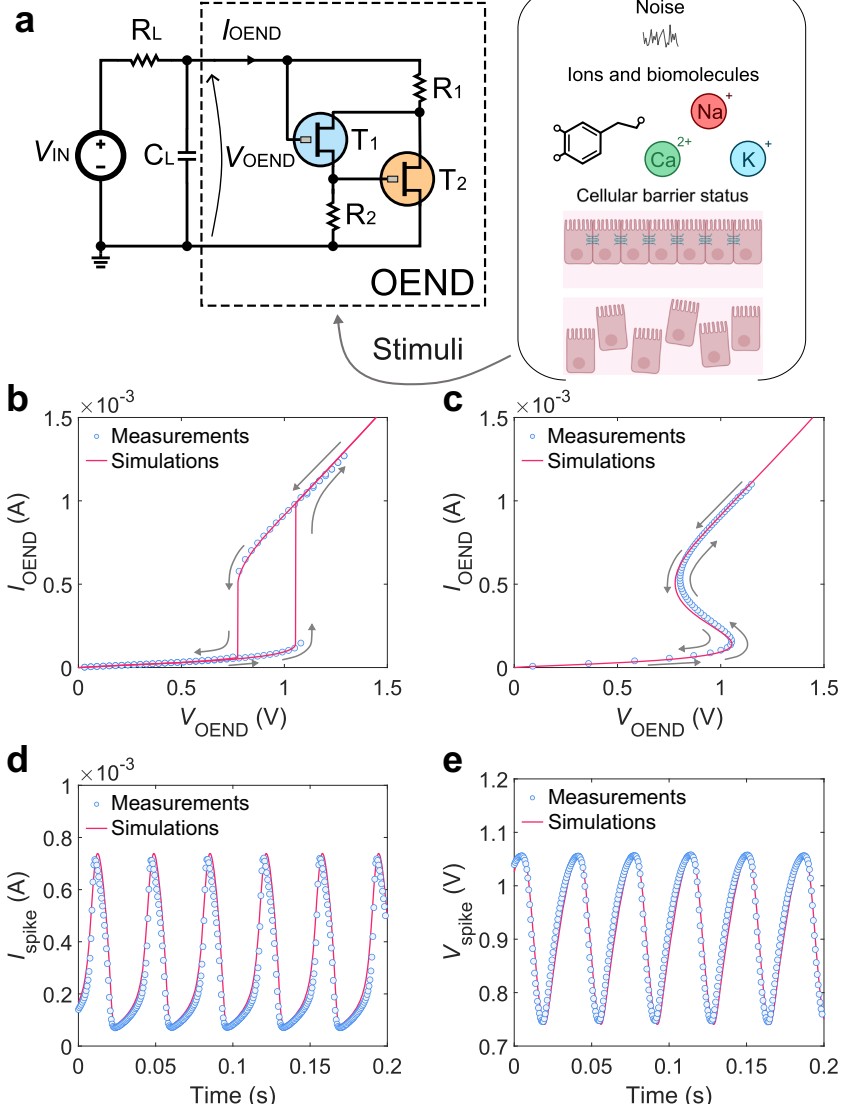

**Fig. 1 | Organic artificial neuron (OAN). a** Circuit diagram of the OAN highlighting the organic electrochemical non-linear device (OEND). $V_{IN}$ is the load voltage generator, $R_L$ represents the load resistor, $C_L$ represents the load capacitance, $T_1$ is the normally on OECT, $T_2$ is the normally off OECT, and $R_1$ and $R_2$ are the two resistors of the OEND. **b** Measured (symbols) and simulated (full line) electrical characteristic of the OEND accessed in voltage mode: $V_{OEND}$ is applied and $I_{OEND}$ is measured. Arrows show the direction of $V_{OEND}$ sweep. **c** Measured (symbols) and simulated (full line) electrical characteristic of the OEND accessed in current mode: $I_{OEND}$ is applied and $V_{OEND}$ is measured. Arrows show the direction of $I_{OEND}$ sweep. **d** Measured (symbols) and simulated (full line) spiking current ($I_{spike}$) of the OAN. **e** Measured (symbols) and simulated (full line) spiking voltage ($V_{spike}$). $T_1$ is a p-type depletion-mode OECTs based on PEDOT:PSS, and $T_2$ is a p-type accumulation mode OECT based on p(g2T-TT).

clarity, the operating regions of the OECTs are also highlighted. During the forward sweep of $V_{OEND}$ from 0 V to positive voltages lower than $V_{ON}$, the current $I_{OEND}$ flows through the branch $R_1$-$T_1$-$R_2$ (region 1, Fig. 2a). This occurs because $T_1$ is a depletion-mode p-type OECT ($V_{TH1} > 0$ V) and $T_2$ is an accumulation mode p-type OECT ($V_{TH2} < 0$ V). Specifically, at small $V_{OEND}$, $T_1$ operates in the linear region, resulting in a small channel resistance $R_{T1}$ and consequently a small source-drain voltage $V_{SD1} = R_{T1}I_{OEND}$ is obtained. The circuit topology dictates that $V_{SD1} = V_{SG2}$, and hence $V_{SG2} < |V_{TH2}|$ (Fig. 2d). During the forward sweep of $V_{OEND}$, as long as $V_{OEND} < V_{ON}$ (region 1, Fig. 2a), $T_1$ remains ON while $T_2$ remains OFF, causing $I_{OEND}$ to flow through the branch $R_1$-$T_1$-$R_2$, resulting in a slope of the $I_{OEND}(V_{OEND})$ characteristic of d$I_{OEND}$/d$V_{OEND} = 1/(R_1 + R_{T1} + R_2)$.

As $V_{OEND}$ increases, the source-drain voltage $T_1$ ($V_{SD1}$) also increases. Once $V_{SD1} > |V_{TH2}|$, $T_2$ turns ON and operates in the saturation region (Fig. 2d). By design, when $T_2$ is ON, its channel resistance

($R_{T2}$) becomes much smaller than $R_2$, causing $I_{OEND}$ to predominantly flow through the branch $R_1$-$T_2$, leading to a sharp increase in current (region 2 in Fig. 2a). This non-linear current enhancement results in a substantial voltage drop across $R_1$. Being $V_{R1} = V_{GS1}$ (Fig. 2c), the overdrive voltage of $T_1$ decreases and eventually $V_{GS1} > V_{TH1}$ causes $T_1$ to turn OFF (Fig. 2c). The voltage required to turn ON $T_2$ is referred to as $V_{ON}$. With further increase in $V_{OEND}$ beyond $V_{ON}$ (region 3 in Fig. 2a), $I_{OEND}$ flows through the branch $R_1$-$T_2$, with a linear increase characterized by a slope d$I_{OEND}$/d$V_{OEND} = 1/(R_1 + R_{T2})$.

When $V_{OEND}$ is swept back to lower voltages, $I_{OEND}$ linearly decreases, resulting in a decrease in the voltage $V_{GS1} = R_1I_{OEND}$. When $V_{GS1} < V_{TH1}$, $T_1$ switches ON in the linear region. Under this condition, $R_{T1}$ is small, and $V_{SD1} = R_{T1}I_{OEND}$ becomes small as well. Given that $V_{SD1} = V_{SG2}$, $T_2$ turns OFF when $V_{SG2} < |V_{TH2}|$, causing the OEND current to sharply decrease (region 4 in Fig. 2a). The voltage required to turn OFF $T_2$ is referred to as $V_{OFF}$. Thus, the OEND exhibits a hysteretic

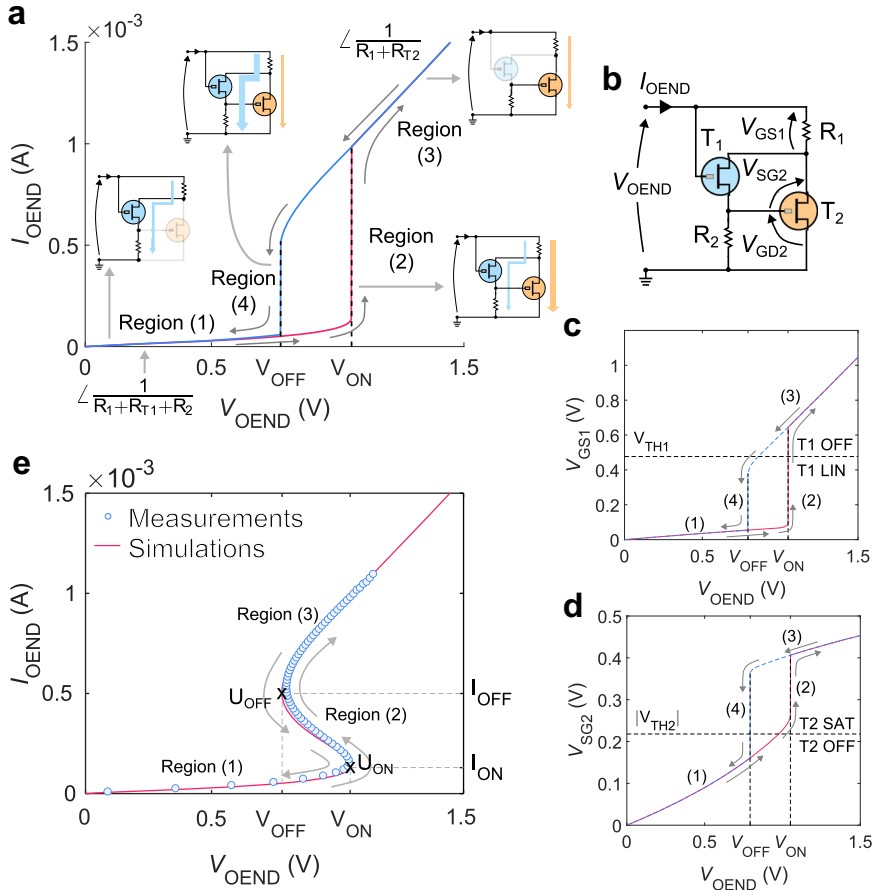

**Fig. 2 | OEND operation. a** OEND electrical characteristic accessed in voltage mode. A voltage ramp ($V_{OEND}$) is applied forward (red line) and backward (blue line), and the current $I_{OEND}$ is measured. The four operating regions and the relevant circuit components in each region of operation are highlighted. **b** OEND circuit showing the internal voltages. **c** Gate-source voltage of transistor $T_1$ ($V_{GS1}$) as a function of $V_{OEND}$. $T_1$ threshold voltage ($V_{TH1}$) is displayed. If $V_{GS1} \geq V_{TH1}$, $T_1$ is turned OFF while if $V_{GS1} < V_{TH1}$, $T_1$ is operated in a linear regime. Forward sweep (red line) and backward sweep (dashed blue line). The numbers refer to the four characteristic due to the non-linear current switching from branch $R_1$-$T_1$-$R_2$ to branch $R_1$-$T_2$ in the forward voltage sweep and from branch $R_1$-$T_2$ to branch $R_1$-$T_1$-$R_2$ in the backward voltage sweep.

operating regions of the OEND, as highlighted in (**a**). **d** Source-gate voltage of transistor $T_2$ ($V_{SG2}$) as a function of $V_{OEND}$. If $V_{SG2} \leq |V_{TH2}|$, $T_2$ is in the OFF state and, if $V_{SG2} > |V_{TH2}|$ results that $T_2$ is in saturation regime. **e** OEND characteristic accessed in current mode calculated with numerical simulations (line) and measured (symbols). A current ramp ($I_{OEND}$) is applied forward and backward, and the voltage $V_{OEND}$ is measured. Forward and backward voltages are overlapped. The points $U_{ON} = (V_{ON}, I_{ON})$ and $U_{OFF} = (V_{OFF}, I_{OFF})$ define the negative resistance region (NRD).

It is important to note that the hysteresis is observed under steady-state conditions (DC operation) as it is inherently related to the OEND circuit configuration, ensuring that $V_{ON}$ and $V_{OFF}$ occur at different voltages. The analytical expressions of $V_{ON}$ and $V_{OFF}$ as a function of material, geometrical, and device parameters are derived in Supplementary Note 2 and are as follows:

$$V_{ON} \cong (R_1 + R_2) g_{m1} \frac{V_{TH2}(2V_{TH1} + V_{TH2})}{2 g_{m1} R_1 V_{TH2} - 1} - V_{TH2} \qquad (1)$$

$$V_{OFF} \cong V_{TH1} - V_{TH2} + \sqrt[\gamma^2]{\frac{V_{TH1}}{g_{m2} R_1}} \qquad (2)$$

where $g_{m1} = (W_1/L_1) t_1 \mu_1 C_{V1}/2$ is the transconductance of the OECT $T_1$ normalized to $V_{SD1}$, $g_{m2}$ the normalized transconductance of the OECT $T_2$, and $W_1, L_1, t_1, \mu_1, C_{V1}$ is the width, length, thickness, mobility, and volumetric capacitance of $T_1$, respectively. As depicted in Supplementary Fig. 3 the amplitude of the OAN voltage oscillations can be calculated as $A_{Vspike} = V_{ON} - V_{OFF}$. This information can be obtained

from the OEND characteristic assessed in voltage mode $I_{OEND}(V_{OEND})$. The operation of the OAN necessitates that $V_{ON} > 0$ V, $V_{OFF} > 0$ V, and $V_{ON} > V_{OFF}$. Equations (1) and (2) demonstrate that these conditions are satisfied when $V_{TH1} > 0$ V, $V_{TH2} < 0$ V, and $V_{TH1} > |V_{TH2}|$. Consequently, $T_1$ has to be a depletion-mode OECT, while $T_2$ has to be an accumulation mode OECT. To further validate the operational parameters of $T_1$ and $T_2$ required for the development of a functional OAN, Supplementary Fig. 4 illustrates that the absence of hysteresis in the $I_{OEND}(V_{OEND})$ characteristic, mirrored by the negative resistance region in the $V_{OEND}(I_{OEND})$ characteristic, occurs in the limiting case $V_{TH1} = |V_{TH2}|$.

When the OEND is operated in current mode, $I_{OEND}$ acts as the independent input variable: $I_{OEND}$ undergoes forward and backward sweeps while the voltage $V_{OEND}$ is recorded. In Fig. 2e the measured data (symbols) and simulated results (line) for the $V_{OEND}(I_{OEND})$ characteristic are presented. The simulations nicely predict the measurements and, unlike the $I_{OEND}(V_{OEND})$ characteristic, the $V_{OEND}(I_{OEND})$ characteristic is non-hysteretic and exhibits S-shape negative differential resistance. The S-NDR behavior is crucial for achieving the spiking behavior of the OAN and can be comprehended based on previous analyses. More in detail, at low input currents, $I_{OEND}$ primarily flows through the left branch $R_1$-$T_1$-$R_2$ causing $V_{OEND}$ to increase almost linearly with the current (Region 1 in Fig. 2e). When $V_{OEND} = V_{ON}$, $T_2$

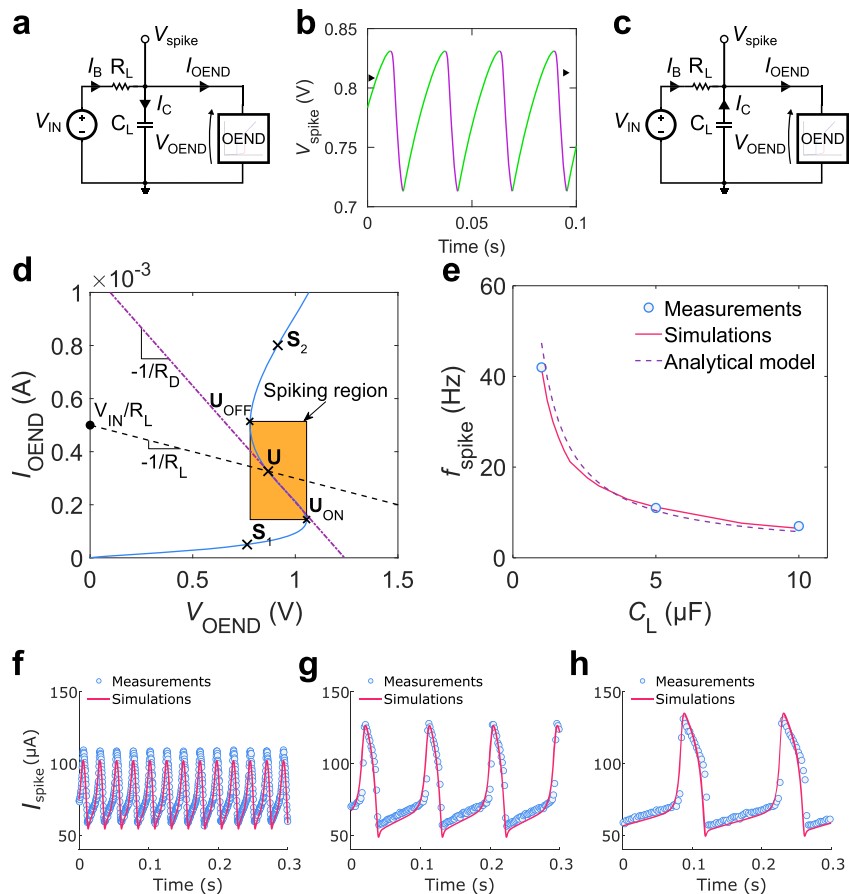

**Fig. 3 | OAN operation and spiking. a** OAN circuit highlighting the current partition when $V_{spike}$ increases, and the capacitor is charged. $V_{IN}$ is the load voltage generator, $R_L$ represents the load resistor, $C_L$ represents the load capacitance. $I_C$ represents the current of the capacitor $C_L$, while $I_B$ is the current flowing through $R_L$. **b** Voltage oscillations ($V_{spike}$) as a function of time. **c** OAN circuit highlighting the current partition when $V_{spike}$ decreases, and the capacitor is discharged. **d** OEND (blue full line) and load-line (dashed line) characteristics. When the load-line characteristic crosses the OEND characteristic in the negative differential resistance region (e.g., point **U**), its response bifurcates, producing voltage and current oscillations. The OEND negative differential resistance is highlighted by dot-dashed purple line. The spiking region is defined by the upper and lower points $U_{ON} = (V_{ON}, I_{ON})$ and $U_{OFF} = (V_{OFF}, I_{OFF})$, respectively. $S_1$ and $S_2$ are two points where the OEND characteristic shows a positive resistance. **e** Spiking frequency $f_{spike}$ as a function of load capacitor ($C_L$). Symbols are the measurements, full line is calculated with the numerical simulations and dashed line is calculated with the analytical model. **f** Spiking current as a function of time measured (symbols) and calculated with non-linear transient simulations (full line) in the case $C_L = 1\,\mu F$, **g** $C_L = 5\,\mu F$, and **h** $C_L = 10\,\mu F$.

turns ON, $I_{OEND}$ can also flow in the right branch $R_1$-$T_2$, leading to a lower voltage drop $V_{OEND}$ across the OEND and resulting in a negative differential resistance (Region 2 in Fig. 2e). With further increases in $I_{OEND}$, the overdrive voltage on $T_2$ rises ($V_{SG2} = V_{SD1}$), causing the channel resistance $R_{T2}$ to decrease and consequently reducing $V_{OEND}$ until $V_{R1} = I_{OEND}R_1$ becomes sufficiently large to deactivate $T_1$. Subsequently, current flows solely through the $R_1$-$T_2$ branch, causing $V_{OEND}$ to monotonically increase with $I_{OEND}$ and restoring positive resistance (Region 3 in Fig. 2e). The non-linear partitioning of current between the two branches of the OEND explains the S-shaped non-linear characteristic and the currents at the switching points of the OEND are determined (refer to Supplementary Note 2):

$$I_{ON} \cong g_{m1} \frac{V_{TH2}(2V_{TH1} + V_{TH2})}{2g_{m1}R_1 V_{TH2} - 1} \qquad (3)$$

$$I_{OFF} \cong \frac{V_{TH1}}{R_1} \qquad (4)$$

The inflection points $U_{ON} = (V_{ON}, I_{ON})$ and $U_{OFF} = (V_{OFF}, I_{OFF})$ (Fig. 2e) given by Eqs. (1–4) define the beginning and the end of the negative resistance region of the OEND and, as will be shown in the

next section, are associated with the excitation and inhibition behavior of the OAN.

## Organic artificial neuron operation and spiking

The spiking activity of the OAN arises from the coupling between the OEND and the biasing network, as depicted in Fig. 3a. When $V_{OEND} < V_{IN}$, the bias current $I_B = (V_{IN} - V_{OEND})/R_L$ charges the capacitor $C_L$, causing the voltage across the capacitor ($V_{spike}$) to increase (Fig. 3b, green line). Since the OAN topology gives $V_{spike} = V_{OEND}$, as $V_{OEND}$ increases and the OEND operates in the negative resistance region, the current $I_{OEND}$ decreases, allowing a larger fraction of current $I_B$ ($I_B = I_C - I_{OEND}$) to charge $C_L$. This further increases $V_{OEND}$ and when $V_{OEND} \geq V_{ON}$ the OEND current significantly increases, reaching the condition $I_{OEND} > I_B$ (Fig. 3c). Subsequently, $C_L$ is discharged and $V_{OEND}$ nonlinearly decreases (Fig. 3b, pink line). When $V_{OEND} \leq V_{OFF}$, $I_{OEND}$ significantly decreases, and when $I_{OEND} < I_B$, $C_L$ is charged again. Therefore, the charging and discharging of the load capacitor depends on the non-linear characteristic of the OEND, input voltage $V_{IN}$ (a DC voltage), and load resistor. The asymmetrical profile of the spikes is a result of the hysteretic profile of the OEND. The connection of the two blocks (i.e., OEND and biasing network) are analyzed in Fig. 3d accounting for the OEND (full blue line) and the biasing network (dashed black line) characteristics. Specifically, when the load-line characteristic crosses

the OEND characteristic in the negative resistance region (e.g., point **U** in Fig. 3d), its response bifurcates, producing voltage and current oscillations.

Figure 3d highlights that the spiking region (orange area) is defined by the points $\mathbf{U_{ON}} = (V_{ON}, J_{ON})$ and $\mathbf{U_{OFF}} = (V_{OFF}, J_{OFF})$, respectively, which can be analytically calculated with Eqs. (1–4). Therefore, OAN spiking activity is inhibited when a stimulus causes the crossing point to shift above $\mathbf{U_{OFF}}$. Conversely, OAN spiking activity is triggered when a stimulus causes the crossing point to shift from below to above $\mathbf{U_{ON}}$, and the distance between the initial position of the crossing point and $\mathbf{U_{ON}}$ defines the OAN excitability (i.e., excitation threshold). Importantly, we note that a unique crossing point in the spiking region is required to avoid the simultaneous presence of non-spiking and spiking states. This condition is fulfilled when the OEND negative resistance $R_D < R_L$ (see the dot-dashed violet line and dashed black line in Fig. 3d). When the load line crosses the OEND characteristic in the positive resistance regions (e.g., point $\mathbf{S_1}$ or $\mathbf{S_2}$, Fig. 3d), stable biasing conditions (i.e., no spiking) are obtained. Indeed, in the bias point $\mathbf{S_1}$ the OEND circuit reduces to a resistor $R_{OEND(\mathbf{S_1})} = R_1 + R_{T1} + R_2$ that, according with the OAN architecture, is connected in parallel to $C_L$ and in series to $R_L$. The voltage across $C_L$ is equal to $V_{OAN} = V_{IN} R_{OEND(\mathbf{S_1})}/(R_{OEND(\mathbf{S_1})} + R_L)$ and therefore no oscillations are obtained. Analogously, in the bias point $\mathbf{S_2}$ the OEND behaves as a resistor $R_{OEND(\mathbf{S_2})} = R_1 + R_{T2}$, and once again a constant output voltage $V_{OAN} = V_{IN} R_{OEND(\mathbf{S_2})}/(R_{OEND(\mathbf{S_2})} + R_L)$ is obtained. Interestingly, the bias points $\mathbf{S_1}$ and $\mathbf{S_2}$ provide a bistability condition of the OAN.

Focusing on the OAN spiking condition, $C_L$ is charged and discharged by a transient current $I_C = (V_{IN} - V_{OEND})/R_L - I_{OEND}$, and integrating over the voltage-current loop the capacitor current equation $I_C = C_L dV_C/dt$, where $V_C = V_{OEND}$ and $t$ is the time, the spiking frequency of the OAN can be calculated as:

$$f_{spike} = \frac{1}{R_L C_L} \left[ \int_{V_{OFF}}^{V_{ON}} \frac{dV}{V_{IN} - V - R_L I_{spike1}(V)} + \int_{V_{ON}}^{V_{OFF}} \frac{dV}{V_{IN} - V - R_L I_{spike2}(V)} \right]^{-1} \quad (5)$$

where $I_{spike1} \approx \frac{V}{R_1 + R_2 + R_{T1}}$ and $I_{spike2} \approx \frac{V + V_{TH2}}{R_1 + R_{T2}}$, is the OEND current during the charging and discharging of $C_L$, respectively. For the sake of clarity, a detailed derivation of Eq. (5) and its analytical solution providing an explicit approximate expression of $f_{spike}$ as a function of the various parameters is reported in Supplementary Note 3. Figure 3e shows the measured (symbols) $f_{spike}$ as a function of $C_L$. The comparison of the measurements with numerical simulations (Fig. 3e, full line) and analytical model given by Eq. (5) (Fig. 3e, dashed line) show that the OAN spiking frequency is accurately predicted by the non-linear transient simulations of the OAN and nicely estimated by the computationally inexpensive analytical model. Interestingly, Eq. (5) accounts for the influence of $R_L$ and $C_L$ (as expected), and quantifies the (less apparent) impact of the material and device parameters through the quantities $V_{ON}$, $V_{OFF}$, $V_{IN}$, and $I_{OEND}$. As a further confirmation, the measured (symbols) and simulated (lines) spiking current as a function of time is displayed in Fig. 3f–h by varying $C_L$. OAN simulations accurately predict the amplitude, frequency, and shape of the measured spiking current. We note that the spiking frequency is a crucial aspect of neural activity. For instance, in sensory systems, the firing rate of neurons can represent various attributes of a stimulus, such as its intensity, duration, or location. Moreover, the spiking frequency of motor neurons is essential for controlling movement. The rate and pattern of spikes in these neurons determine the strength and timing of muscle contractions. Every class on neuron has its own characteristic range of firing frequency, while deviations might be an indication of pathological conditions. Therefore, it is essential to control the firing frequency of the OANs by design. The developed numerical framework serves as a valuable tool for the design of OANs and Eq. (5) and its analytical formulation given by Supplementary Eq. (26) offers a valuable model for the

rational design of $f_{spike}$ as a function of the OAN material, geometrical, physical and device parameters.

To gain more insight on the impact of the various material, geometrical, and device parameters on the OAN performance, we take advantage of numerical simulations. Specifically, we performed parametric analysis of the OAN performance as a function of threshold voltage ($V_{TH1}$, $V_{TH2}$), transconductance ($g_{m1}$, $g_{m2}$), volumetric capacitance ($C_{V1}$, $C_{V2}$), width ($W_1$, $W_2$), length ($L_1$, $L_2$), thickness ($t_1$, $t_2$), and resistance ($R_1$, $R_2$). To assess the impact of each parameter on the OAN performance, we modified one parameter at a time while keeping all other parameters constant. The parameters are varied accounting for the largest range that ensures OAN oscillation. The corresponding $I_{OEND}$-$V_{OEND}$ characteristics accessed in the current mode are displayed in Supplementary Figs. 6, 7, and 8. Numerical simulations in Fig. 4a–h show that $f_{spike}$ can be modulated in the range 10–100 Hz by tuning the parameters $V_{TH1}$ (Fig. 4a), $V_{TH2}$ (Fig. 4b), $g_{m1}$ (Fig. 4c), $C_{V1}$ (Fig. 4e), $C_{V2}$ (Fig. 4f), and $R_2$ (Fig. 4h) while it is almost insensitive to $g_{m2}$, (Fig. 4d) and $R_1$. (Fig. 4g). More in detail, $f_{spike}$ increases by reducing $V_{TH1}$, $g_{m1}$, $C_{V1}$, and $R_2$, while an opposite trend is obtained with $V_{TH2}$. The relation between $f_{spike}$ and the material parameters can be linked to the OEND characteristic. $f_{spike}$ can be enhanced by minimizing the current flowing in the branch $R_1$-$T_1$-$R_2$ and maximizing the current flowing in the branch $R_1$-$T_2$. Interestingly, $C_{V1}$ (Fig. 4e) and $C_{V2}$ (Fig. 4f) are very significant material parameters influencing $f_{spike}$, with the spiking frequency being maximum when $C_{V1}$ and $C_{V2}$ are minimized. To further explore this aspect, Supplementary Fig. 9 analyzes the OAN spiking frequency while also considering the relationship between $C_{V1}$, $C_{V2}$, and load capacitor $C_L$. We systematically varied both $C_{V1}$ and $C_{V2}$, and the minimum $C_L$ required for OAN spiking is calculated. Supplementary Fig. 9a reveals that the minimum $C_L$ amounts to $10^{-7}$ F and is achieved when $C_{V2}$ falls in the range 50–100 F cm$^{-3}$ and $C_{V1}$ is approximately 50 F cm$^{-3}$. Supplementary Fig. 9b highlights that the maximum spiking frequency, $f_{spike} = 150$ Hz, is attained when both $C_{V2}$ and $C_L$ are minimized. We observe that minimizing $C_{V2}$ leads to a reduction in the capacitance of OECT $T_2$, which can also be achieved by adjusting the geometrical parameters. Supplementary Fig. 10 demonstrates that $f_{spike}$ is enhanced when $W_2$, $L_2$, and $t_2$ are minimized. Faster spiking frequencies give rise to a smaller amplitude of oscillations and vice-versa, as confirmed in Supplementary Figs. 11–14. As expected, the OAN power consumption ($P_{OAN}$) increases with the spiking frequency (Supplementary Figs. 15 and 16). By contrast, as displayed in Fig. 4 the energy per spike $E_{spike} = P_{OAN}/f_{spike}$ decreases with increasing the spiking frequency, indicating that the OAN efficiency is enhanced at faster oscillations. A minimum energy per spike equal to $1 \times 10^{-6}$ J is obtained by minimizing only $C_{V2}$, while keeping all the other parameters at the nominal value. We note that energy per spike of the OAN is significantly higher than the energy consumed by biological neurons[46,47] and point out that future work should focus on this direction. In Supplementary Figs. 17 and 18, the analysis of the impact of material parameters on the OAN performance is further extended to several OMIECs suitable for depletion-mode (transistor $T_1$) and accumulation mode (transistor $T_2$) devices. The analysis confirms that OMIECs with large volumetric capacitance and low mobility (e.g., p(gNDI-g2T) in Supplementary Fig. 17), provide OANs with very limited spiking frequency, large energy per spike, and low power consumption. This is further confirmed in Supplementary Fig. 18 where the material with the lowest mobility, i.e., p(gBDT-g2T), provides OANs with minimum spiking frequency and maximum power consumption.

## Excitability and noise-induced activity

The ability to generate spikes, named excitability, is a fundamental property of neurons that allows them to transmit and process information within the nervous system. Neurons are classified into different categories based on their level of excitability. The excitability of

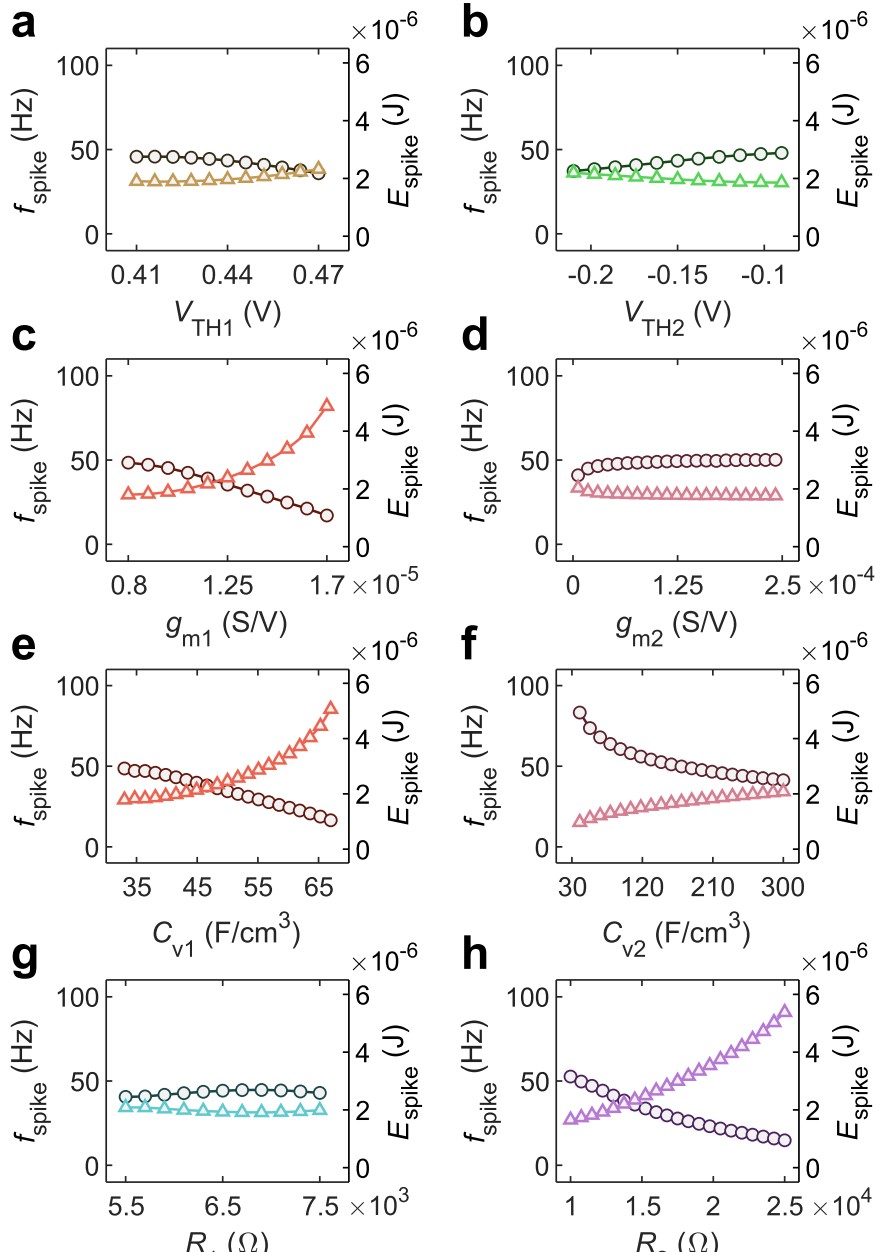

**Fig. 4 | OAN spiking frequency and energy.** Spiking frequency ($f_{spike}$, left axis and circle symbols) and energy per spike ($E_{spike}$, right axis and triangle symbols) as a function of the OAN parameters. $E_{spike} = P_{OAN}/f_{spike}$, where $P_{OAN}$ is the power consumption. $f_{spike}$ and $E_{spike}$ are computed as a function of (**a**) threshold voltage of T$_1$ $V_{TH1}$, **b** threshold voltage of T$_2$ $V_{TH2}$, **c** normalized transconductance of T$_1$ $g_{m1}$, **d** normalized transconductance of T$_2$ $g_{m2}$, **e** volumetric capacitance of T$_1$ $C_{v1}$, **f** volumetric capacitance of T$_2$ $C_{v2}$, **g** resistance $R_1$, and **h** resistance $R_2$.

neurons expresses their tendency to elicit spikes when receiving incoming signals or stimuli. Highly excitable neurons have a low threshold for generating action potentials, and even small changes in their membrane potential can trigger an action potential. Some neurons require stronger or more prolonged stimuli to reach the threshold for firing an action potential. These neurons are less sensitive to small fluctuations in their membrane potential and may have a higher threshold for activation.

Taking advantage of the functionality of the OAN, artificial neurons with varying levels of excitability can be achieved. For this purpose, a sinusoidal excitation signal is injected into the electrolyte medium of T$_1$. As depicted in Fig. 5a, a slight increase of a few millivolts in the potential ($V_{exc}$) of the electrolytic medium, corresponding to the same range as extracellular biopotentials[48], modulates the excitability

of the OAN. During excitation, a phase-locked bursting activity is observed. Furthermore, Fig. 5a demonstrates a high level of agreement between the measurements (gray lines) and numerical simulations (red lines). The numerical simulations reveal that the injected electrolyte voltage $V_E(t)$ gives rise to a time-varying modulation of the OEND-load line crossing point (Fig. 5b) equal to $[V_{IN} + V_{exc} + V_E(t)]/R_L$. When the crossing point is positioned outside the spiking region (Fig. 5c, thick dark purple segment $U(t) < U_{ON}$), a constant bias voltage across C$_L$ is obtained during the whole period of the excitation signal and, consequently, the OAN is silent (Fig. 5a top panel). To probe the OAN excitability, $V_{exc}$ is increased by a few millivolts, and during the timeframe where the injected signal moves the crossing point **U** in the negative resistance region (Fig. 5d, thick bright purple segment $U(t) > U_{ON}$) firing activity is triggered. Excitation can be further

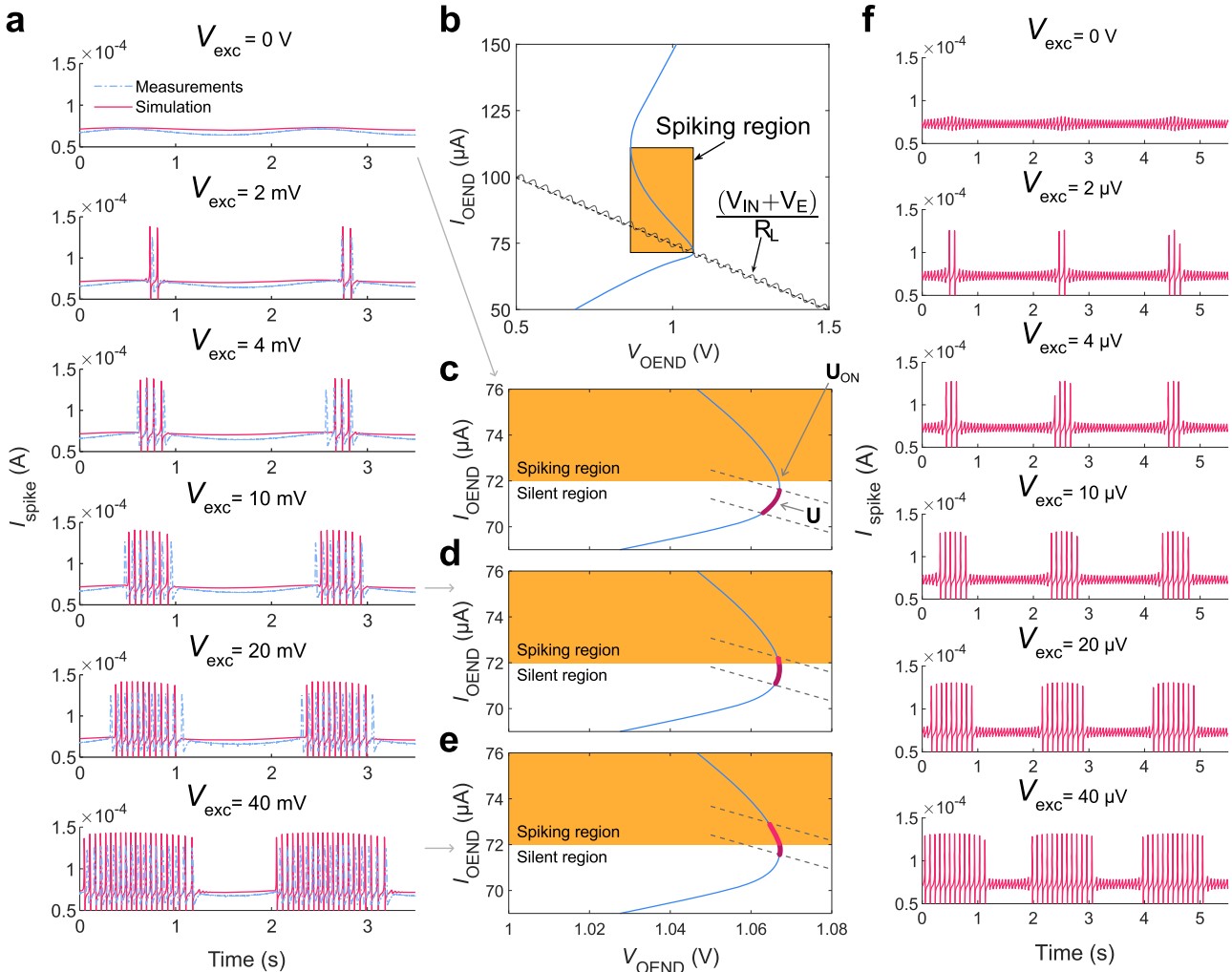

**Fig. 5 | Excitability and phase-locked bursting. a** Measured (dot-dashed lines) and calculated (full line) spiking current as a function of time at various input excitation signals $V_{exc} = 0$–$40$ mV, and $V_E(t)$ is a sinusoidal signal with amplitude 20 mV$_{pp}$ and frequency 5 Hz. **b** Current-voltage characteristics $I_{OEND}(V_{OEND})$ of the OEND (blue full line) and load-line (dashed line) highlighting the impact of the excitation signal. The excitation signal is depicted by a sinusoidal full line superimposed on the load line. **c** Zoom showing the range where the crossing point is modulated by the excitation signal (thick dark purple line). $V_{exc} = 0$ V, the crossing point is in the silent region and OAN is not spiking. **d** $V_{exc} = 10$ mV, the crossing point is modulated by the excitation signal as a function of time. The range where the crossing point is in the spiking region and silent region is marked with a thick brighter and a darker purple line, respectively. **e** $V_{exc} = 40$ mV, the crossing point is in the spiking. The thick bright purple line shows the range where the crossing point is in the spiking. **f** Spiking current as a function of time at various input excitation signals $V_{exc} = 0$–$40\,\mu$V.

enhanced by increasing $V_{exc}$ (Fig. 5e), and the bursting width is phase-locked with the electrolyte signal (Fig. 5a), which emulates a fundamental feature of biological neurons[49].

The degree of excitability can be adjusted to the desired level by appropriately configuring the OAN parameters. To explore this important aspect, we conducted numerical simulations, systematically varying the material and device parameters. To evaluate the impact of each parameter, we varied one parameter at a time while keeping all others constant. Supplementary Fig. 19 depicts the excitation threshold voltage, $V_{exc}$, as a function of OAN parameters. Excitability increases with $g_{m1}, g_{m2}, C_{V1}, C_{V2}, R_1$ and $R_2$ while decreases with $V_{TH1}$ and $V_{TH2}$. These findings suggest that the level of excitability can be finely tuned, ranging from a few microvolts to hundreds of millivolts. This remarkable degree of tunability is achieved by configuring the OAN parameters and allows, for example, phase-locked bursting with excitation input signals as low as a few microvolts. (Fig. 5f).

Modulation of the OAN firing threshold allows to emulate other relevant features of biological neurons, as for example noise-induced activity. Noise, in the context of neurons and neural systems, plays a role in enhancing sensory processing, promoting robustness, generating variability, and shaping network dynamics. Noise couples with neuronal dynamics affecting the neuronal firing properties. To emulate extracellular noise fluctuations, white noise is applied to the electrolyte. The noise-induced transition from tonic to bursting of the OAN activity is shown in Fig. 6. Injected noise sums up to $V_{IN}$ and the OAN crossing point dynamically shifts with the amplitude and time evolution of the noise. In the case of a small amplitude noise (e.g., $V_{pp} = 25$ mV, Fig. 6a), a tonic firing is displayed. Numerical simulations allowed us to calculate the $I_{OEND}$-$V_{OEND}$ characteristic as well as the load-line crossing point as a function of noise. As displayed in Fig. 6b (gray area around the dashed load line) the amplitude of noise is not sufficient to shift the crossing point outside the spiking region. The OAN is resilient to the noise in the liquid environment and therefore regular firing is obtained. By increasing the level of noise (e.g., $V_{pp} = 50$ mV, Fig. 6c), there is a time-varying probability that the crossing point is shifted in the silent region (Fig. 6d) and in correspondence of such stochastic

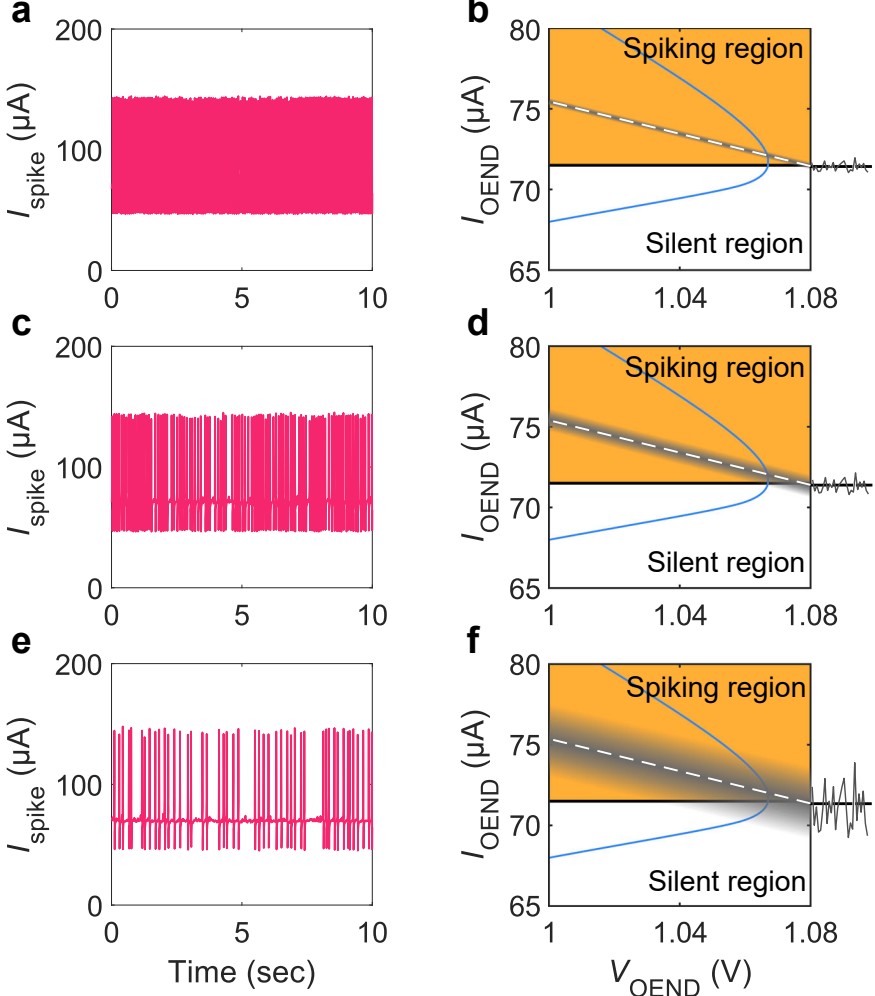

**Fig. 6 | Noise-induced activity.** White noise signal with amplitude $V_{pp}$ is injected into the electrolyte. **a, c, e** Spiking current $I_{spike}$ as a function of time at various amplitudes ($V_{pp}$) of noise signal injected into the electrolyte. **b, d, f** Impact of the noise on the current-voltage characteristics $I_{OEND}(V_{OEND})$ of the OEND (blue full line) and load-line (dashed line) calculated with numerical simulations. The noise modulates the position of the load line as a function of time (shade area) and elicits interrupted spikes. **a, b** $V_{pp} = 25$ mV, **c, d** $V_{pp} = 50$ mV, and **e, f** $V_{pp} = 150$ mV. Random packets of spikes are triggered by noise.

events OAN spiking is suppressed. Further increasing the noise amplitude (e.g., $V_{pp} = 150$ mV, Fig. 6e), the stochastic modulation of the OAN firing threshold is enhanced (Fig. 6f) and random packets of spikes are triggered by noise. Interestingly, as illustrated in Supplementary Fig. 20, it is possible to design OANs with spiking activity induced by noise: OAN is initially silent (below the excitation threshold) and noise can trigger the spiking activity at various intensities – ranging from a few spikes to nearly tonic firing – by increasing its excitability. This shows the high degree of reconfigurability inherent in the OAN.

**Neuromorphic ion sensing**

Ions are fundamental biological and physiological regulators for vital processes in every living organism. Ion regulation in living organisms involves intra and extracellular fluctuations from the resting condition, and small deviations from the optimal equilibrium levels can be associated with pathological states[50]. The intracellular-extracellular physiological concentration in mammalian cells is $3{-}160 \times 10^{-3}$ M for potassium ($K^+$) and $10{-}150 \times 10^{-3}$ M for sodium ($Na^+$) ions[51]. Under resting conditions, neurons maintain a steep gradient between low intracellular free calcium ($Ca^{2+}$) concentration ($0.1{-}0.5 \times 10^{-6}$ M) and high extracellular $Ca^{2+}$ levels ($\sim 1 \times 10^{-3}$ M)[52]. In biological neurons, the changes in ion concentration between the

intracellular and extracellular medium modulate their excitability and firing threshold[53].

The OAN, operating in a liquid environment, displays ion-concentration-dependent spiking properties. In biointerfacing scenarios involving biological cells and OANs, the shared electrolyte serves as a common extracellular space. Information exchange and signaling, such as ionic species transmission, between the biological and artificial domains occur through this shared extracellular medium. Figure 7a shows the measured (symbols) OAN output current ($I_{spike}$) as a function of time by varying the ion concentration in the external medium, i.e., the artificial extracellular space. The OAN excitability and spiking frequency are controlled by the ion concentration of the electrolytic medium. More in detail, at $Na^+$ concentrations below the physiological limit, oscillations are suppressed. Increasing the ion concentration firing is triggered and when the extracellular concentration is varied in the physiological range, $f_{spike}$ is modulated from 20 Hz to 50 Hz. The measurements (symbols, Fig. 7a) are accurately predicted by the simulations (solid lines) in the whole range of ion concentration, showing that the numerical framework adequately accounts for the ionic–electronic interaction in the liquid environment. Then, as detailed in Supplementary Note 1, the impact of the ion concentration is accounted for. The spiking frequency as a function of $Na^+$ calculated with the

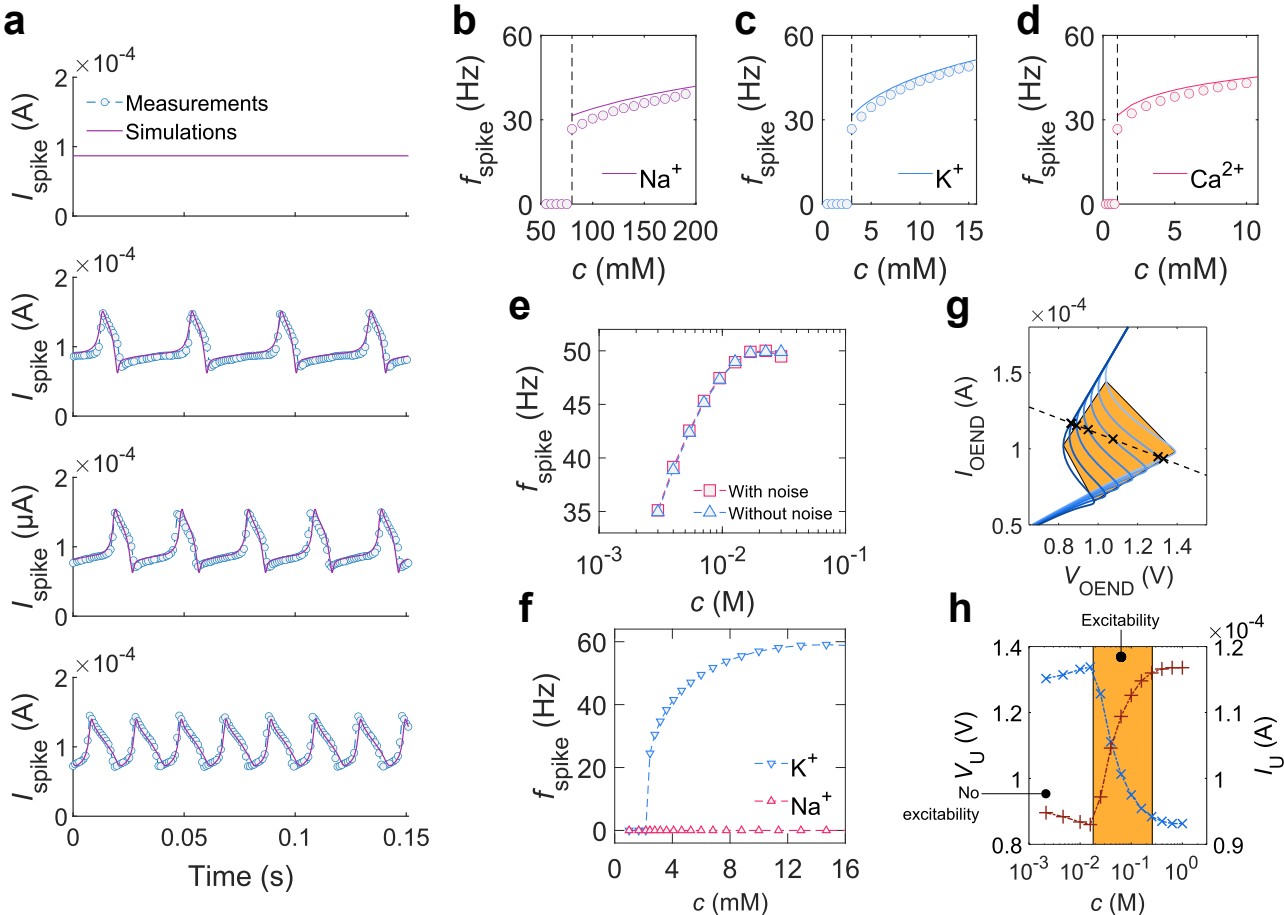

**Fig. 7 | Neuromorphic ion sensing. a** Measured (symbols) and modeled (lines) spiking current $I_{spike}$ as a function of time. The OAN is exposed to various $Na^+$ concentrations. When the ion concentration is below the minimum physiological level, no spiking is observed. When the ion concentration is above the excitation threshold, OAN spiking is obtained and its spiking frequency is modulated with $c_{Na^+}$. **b** Spiking frequency $f_{spike}$ as a function of $Na^+$ concentration. Symbols are the numerical simulations and the full line is calculated with the analytical model in Supplementary Eq. (26). **c** $f_{spike}$ as a function of $K^+$ concentration. **d** $f_{spike}$ as a function of $Ca^{2+}$ concentration. In **b**–**d** the vertical dashed line shows the minimum physiological concentration of each ion. OAN spiking is triggered by the specific ion concentration. **e** Spiking frequency as a function of ion concentration without and with noise. **f** Ion-selective spiking: spiking frequency as a function of selected $K^+$ concentration. The OAN is not spiking when the concentration of interfering ions ($Na^+$) is varied. **g** OEND characteristics $V_{OEND}(I_{OEND})$ modulated by the ion concentration (solid lines). Load line (dashed dark line) crosses the OEND characteristics in various points inside the spiking region (orange area), depending on ion concentration. At large ion concentration, the crossing point $\mathbf{U} = (V_{\mathbf{U}}, I_{\mathbf{U}})$ is outside the spiking region and the OAN is silent. **h** Crossing voltage $V_{\mathbf{U}}$ (cross symbols) and current $I_{\mathbf{U}}$ (plus symbols) as a function of ionic concentration $c$. $V_{\mathbf{U}}$ decreases with $c$ while $I_{\mathbf{U}}$ increases. In the OAN excitability region (orange region) both $V_{\mathbf{U}}$ and $I_{\mathbf{U}}$ logarithmically increase with ion concentration. Outside the excitability region, the OAN is silent.

analytical model (Supplementary Note 3) is displayed in Fig. 7b. The results obtained with the computationally inexpensive analytical model are in very good agreement with both the measurements and numerical simulations. The model reveals that the modulation of the spiking frequency is inherently related to the electrochemical properties of the OECTs connected in the OAN configuration. More in detail, the potential drop at the gate/electrolyte interface and the Donnan's potential at the electrolyte/polymer interface[40] are mirrored in a variation of the threshold voltage of $T_1$ ($V_{TH1}$). According to Eqs. (1) and (2), $V_{TH1}$ modulates the $V_{ON}$ and $V_{OFF}$ voltages, which set the maximum charging and discharging voltage of $C_L$, respectively, and eventually define the integration domain in Eq. (5), viz. $f_{spike}$. As displayed in Supplementary Fig. 21, the $I_D$–$V_G$ characteristics of the OECT systematically shift to more negative voltages with increasing ion concentrations. More negative $V_{TH1}$ reduces both $V_{ON}$ and $V_{OFF}$, thus resulting in a faster spiking activity. In Fig. 7c, d the analysis is extended to other ions, namely $K^+$ and $Ca^{2+}$, confirming the excitability, firing threshold, and frequency-modulation features of the OAN. Supplementary Figs. 22 and 23 show the corresponding spiking current as a function of time. A modulation of the spiking

frequency equal to 38.95 Hz dec$^{-1}$ and 26.69 Hz dec$^{-1}$ is obtained with monovalent and divalent ions, respectively, which correspond to a frequency modulation of about $\Delta f_{spike}/f_{spike} = 134\,\%$ for $K^+$ and $Na^+$, and $\Delta f_{spike}/f_{spike} = 67\%$ in the case of $Ca^{2+}$.

A key distinctive capability of neuromorphic circuits is their intrinsic robustness against interfering signals. To this aim, noise is intentionally injected into the electrolyte of the OAN to observe its effect on the spiking behavior at different ion concentrations (Supplementary Fig. 24). While the amplitude of the spiking signal is minimally influenced by noise, the information encoded in the spiking frequency remains unchanged. This is evident in Fig. 7e, where the spiking frequency as a function of ion concentration shows nearly identical curves regardless of the presence of noise. To further illustrate this point, we compare our neuromorphic ion-sensing approach with conventional methods based on OECTs[54–56]. When noise of the same amplitude and spectral characteristics as that used in the OAN case is injected into the electrolyte of the OECT, significant differences emerge as shown in the transfer characteristics ($I_D$-$V_G$) as a function of ion concentration (Supplementary Fig. 25). Unlike the OAN, the OECT drain current is substantially impacted by noise,

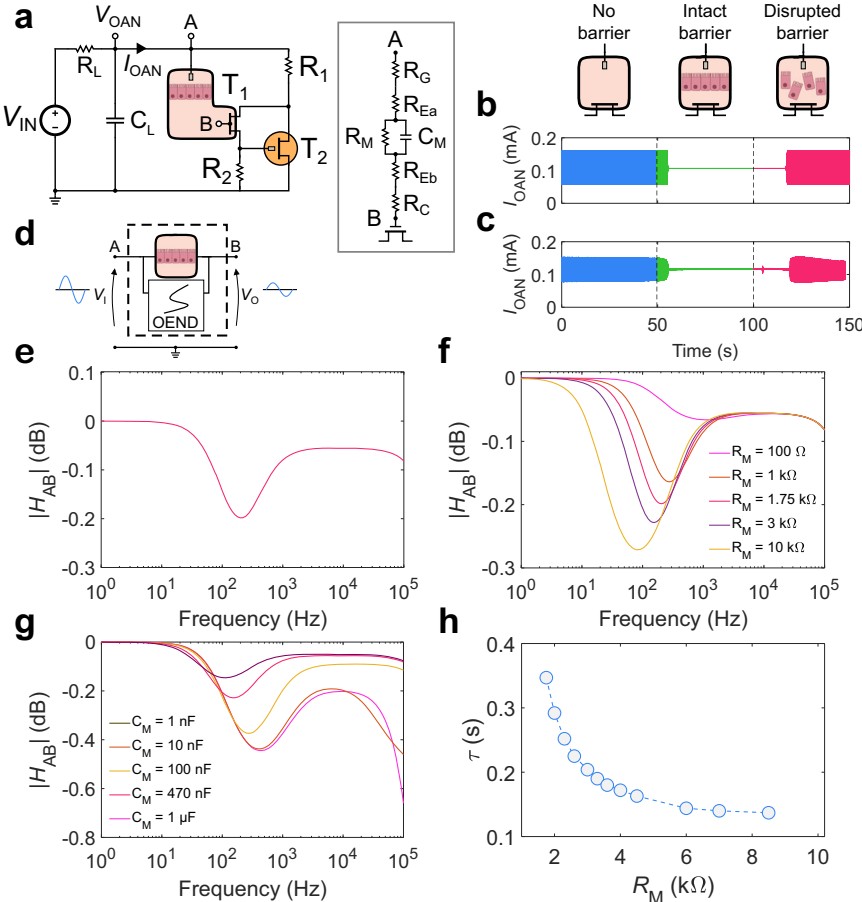

**Fig. 8 | In-liquid biointerfacing. a** OAN circuit including the cellular barrier between the apical electrolyte (point A) and the basal electrolyte (point B). The model of the cellular barrier includes the gate $R_G$ resistance, the apical electrolyte equivalent resistance $R_{Ea}$, and the basal electrolyte equivalent resistance $R_{Eb}$, the ion resistance $R_M$ and capacitance $C_M$ of the membrane, and the channel resistance $R_C$. **b** Measurements of the OAN without cellular barrier (blue line), with intact cellular barrier interfaced with the OAN (green line), and with disrupted cellular barrier (purple line). Insertion and disruption of the cellular barrier is highlighted with the vertical dashed lines. **c** Transient numerical simulations of the OAN without cellular barrier (blue line), with intact cellular barrier interfaced with the OAN

(green line), and with disrupted cellular barrier (purple line). Insertion and disruption of the cellular barrier is highlighted with the vertical dashed lines. **d** Model of the OAN and cellular barrier accounting for an input signal at the apical electrolyte (point A) and output signal at the basal electrolyte (point B). **e** Calculated transfer function module between A-B $|H_{AB}|$ for the frequency range 1 Hz–$10^5$ Hz. The signal is attenuated in a frequency range relevant to the oscillation (range 10–150 Hz). **f** Impact of the membrane resistance on the OAN oscillations. **g** Impact of the membrane capacitance on the OAN oscillations. **h** Time constant of the oscillation decay ($\tau$) after the cellular barrier insertion as a function of $R_M$.

leading to corruption of information regarding ion concentration. This comparison underscores the intrinsic robustness of neuromorphic approaches, which encode information in the frequency domain, over conventional methods.

Adding another layer of biophysical realism to the OAN response involves integrating selectivity to specific ions, thus obtaining ion-selective excitation and spiking. OANs capable of exhibiting ion-specific oscillatory activities are realized by integrating ionophore-based selective membranes[57] (ISM) at the interface between the channel and electrolyte of $T_1$. The ISM generates a voltage at the membrane/electrolyte interface ($V_{ISM}$) in response to the specific ion concentration within the electrolyte (Supplementary Note 1). The integration of ISM in the OAN architecture results in a variation of the OECT $T_1$ threshold voltage $V_{TH1}$ (Supplementary Fig. 26). Being the OAN excitability dependent on $V_{TH1}$ (Supplementary Fig. 19) variation of the target ion concentration results in ion-selective excitability and spiking of the OAN. Figure 7f shows that a $K^+$ selective OAN exhibits oscillations when immersed in a KCl electrolyte, with the frequency of oscillations increasing with the concentration of the selected ion type. As a control experiment, $Na^+$ is used as interfering ions and the OAN is silent (Fig. 7f),

demonstrating ion-selective spiking activity. Analogously, a $Na^+$ selective OAN exhibits oscillations when immersed in a NaCl electrolyte and is silent in KCl (Supplementary Fig. 27).

The ion modulation of the OAN spiking features can be rationalized by analyzing the OEND characteristics as a function of the ion concentration. For the sake of clarity, from here on we considered a variation of the ion concentration in the electrolyte of the transistor $T_1$, and analogous considerations hold for $T_2$. The numerical simulations displayed in Fig. 7g show the OEND characteristics calculated as a function of the ion concentration. The $I_{OEND}$-$V_{OEND}$ characteristics are almost perfectly overlapped when the OEND operates in the positive resistance regions because at small currents $I_{OEND} \approx V/(R_1 + R_2)$, at large currents $I_{OEND} \approx V/R_1$ and both $R_1$ and $R_2$ are not affected by the ion concentration. Conversely, the OEND characteristics in the negative differential resistance region are significantly modulated by the ion concentration and the crossing point with the load-line characteristic (dashed line in Fig. 7g) systematically shifts with the ion concentration. Specifically, at small ion concentrations, the operating point of the OAN (i.e., OEND-load-line crossing point) lies in the non-excitable region and the OAN is silent. By increasing the ion concentration, a transition of the crossing

point in the negative resistance region is obtained (orange region) and the OAN is firing. The crossing point consistently shifts to smaller voltages and larger currents with the increasing of the ion concentration, which enables faster charging and discharging of the load capacitor $C_L$ and results in faster OAN oscillations. This behavior is displayed up to a maximum ion concentration where $V_{ON} = V_{OFF}$ (see Supplementary Note 4). Reaching this limit, the OEND characteristic does not show a negative resistance anymore, and therefore the OAN spiking region is lost. The excitability region as a function of the ion concentration is displayed in Fig. 7h. OAN excitability requires a crossing point $(V_U, I_U)$ located in the negative resistance region of the OEND. The OAN bias voltage $V_U$ systematically reduces with the ion concentration, while the bias current $I_U$ increases. According to the previous analysis, larger bias currents and smaller voltages result in enhanced $f_{spike}$ when the OAN is operated in the excitability region. Finally, as a convenient design approach, we found that the minimum ion concentration triggering OAN oscillations can be defined with the load-line parameters while the maximum ion concentration of the excitability region can be defined with the concentration-dependent OEND parameters.

## Biointerfacing

Direct interfacing with biology has been demonstrated by incorporating a biological membrane in the OAN[34]. Biological membranes are cellular barriers that control the transport of ions, small molecules, and nutrients through the separated compartments of tissue, thus regulating essential physiological functions in humans, animals, and plants. The transcellular and paracellular pathways are finely regulated by the cellular barriers and the tight-junction proteins allow intercellular sealing and control the paracellular fluxes. A biohybrid OAN is obtained by integrating a cellular barrier between the channel and the gate of $T_1$. Cell medium is used as an electrolyte and the cellular barrier separates the electrolyte into two compartments. Therefore, there is no physical contact between the membrane and the electrode or the device channel. Coupling between the cellular barrier and the OAN components is obtained by means of the cell medium, which represents a shared extracellular space[34,58,59]. The circuit diagram of the biohybrid OAN is displayed in Fig. 8a. A detailed view of the cellular barrier interface with the transistor $T_1$ is given in Supplementary Fig. 28. We used a non-polarizable Ag/AgCl gate electrode immersed in the apical compartment, which set the potential of the electrolyte to approximately the applied $V_G$[42,60,61]. We also note that, in general, it is not straightforward to compare qualitatively forced (external) voltages with biopotentials, because the actual voltage that a sensing unit (such as the OAN) experiences is also related to the impedance of a voltage source. For instance, in the case of a source of biological activity (a cell), the potential would depend on the cell-to-device impedance. As detailed in Fig. 8a, the gate electrode is modeled as a resistor, $R_G$, in series to the ionic resistance of the electrolyte in the apical compartment, $R_{Ea}$, due to the ion transport from the gate to the cellular barrier in the electrolyte medium of the apical compartment. The cellular barrier is described by a capacitor $C_M$ in parallel to a resistor $R_M$, and the ionic transport in the electrolyte medium of the basal compartment is modeled by the resistor $R_{Eb}$. Electrochemical impedance spectroscopy (EIS) measurements in Supplementary Fig. 29 show that the impedance of intact cellular barrier is much larger than the gate and ionic resistance of apical and basal electrolytes, $|Z_M| \gg R_G + R_{Ea} + R_{Eb}$. Focusing on the cellular barrier model $C_M$ accounts for the ion accumulation at the basal and apical barrier interfaces. $C_M$ is relevant when the TJs are intact since in this condition the ions are not flowing through the barrier. $R_M$ accounts for the ionic transport through the barrier and it is relevant when TJs are disrupted with toxins or transiently open with neuromodulators[58]. The barrier parameters are obtained from EIS measurements (Supplementary Fig. 29). It is

important to observe that the complexity of the equivalent circuit required may vary depending on the types of cells interfacing with the OAN[62]. The OAN spiking current as a function of time and barrier status is displayed in Fig. 8b, c. Oscillations with a spiking frequency of 23 Hz are obtained without the cellular barrier (blue line). Upon barrier insertion the oscillations are damped, and OAN firing activity is completely suppressed after about 2 s (green line, Fig. 8b, c). The addition of toxin compounds in the cellular medium gives rise to the opening of the TJs and barrier disruption eventually results in a restored spiking of the OAN (pink line, Fig. 8b, c). Importantly, Supplementary Fig. 30 shows that the toxic compounds do not affect the device performance.

We analyzed the excitability modulation of the OAN due to the cellular barrier by means of numerical simulations. The $I$-$V$ characteristics of the OEND and load line are displayed in Supplementary Fig. 31. The crossing point between the OEND and load line is located in the spiking region but, in contrast to the previous cases, it is neither affected by the barrier insertion nor by the barrier status. Interestingly, Fig. 8b, c show that the OAN output is damped after the insertion of the barrier (green line). This suggests that the spiking activity is dynamically attenuated by the barrier. To investigate this effect, we performed AC analysis of the OAN. As schematically depicted in Fig. 8d, we injected a small input signal ($v_i$) in the apical electrolyte and the output signal ($v_o$) is obtained in the basal electrolyte. The frequency ($f$) of the input signal is scanned from 1 Hz to $10^5$ Hz and the Laplace transform of the signals provides the transfer function $H_{AB} = V_o(s)/V_i(s)$, where $s = \mathrm{j}2\pi f$. Figure 8e shows that the signal is attenuated ($|H_{AB}| < -0.2$ dB) by the cellular barrier in the range of frequency relevant for the OAN oscillation (10–150 Hz). Therefore, at each cycle, an attenuated spike is fed back to the OAN resulting in an overall dumping of the oscillation. The impact of the membrane parameters $R_M$ and $C_M$ is analyzed in Fig. 8f, g, respectively. Figure 8f shows that spike attenuation takes place when $R_M > 10^3$ Ω and it progressively increases with the barrier resistance. We note that barrier resistance depends on the cellular barrier status. Conversely, Fig. 8g shows that $C_M$ has a minor effect on the magnitude of the spike attenuation and larger $C_M$ extends the frequency range where barrier attenuation is effective. As a result, the excitability of the OAN is controlled by the barrier status (Supplementary Fig. 32) and information on the TJs modulation, mirrored in the barrier resistance, is obtained from the damping time constant $\tau$ of the spiking output (Fig. 8h).

## Discussion

By combining experiments, new numerical tools, and analytical expressions, we unrevealed the operation of organic electrochemical artificial neurons. Starting from the investigation of the OEND, which is the core of the OAN, we systematically rationalized the OAN fundamental operations focusing on spiking frequency, voltage and current amplitude of the output oscillations, power consumption, and energy per spike. Numerical simulations accurately predicting the measurements have shown that such OAN features intimately depend on the material and device parameters. Table 1 highlights the impact of the various material, biochemical, geometrical, and device parameters on the OAN performance, providing general guidelines to optimize and engineer the OAN and to shape the spiking profile. An overview of artificial spiking neurons providing biophysical realism capabilities (viz. emulation of real neuron spiking behavior) is displayed in Table 2. The wide biorealistic repertoire of the OAN including excitability, noise-induced firing activity, neuromorphic ion sensing, and in-liquid bio-interfacing has been rationalized. We have shown that excitability threshold and noise-induced firing activity are inherently due to the organic electrochemical non-linear device and the key design parameters have been identified and discussed. We demonstrated that the OAN excitability and spiking frequency controlled by the ion

**Table 1 | OAN performance as a function of the material and device parameters**

| | Parameter | $A_{Vspike}$ | $A_{Ispike}$ | $V_U$ | $f_{spike} \times A_{Vspike}$ | $P_{OAN}$ | $E_{spike}$ | $\tau_M$ |
|---|---|---|---|---|---|---|---|---|
| Material | $V_{TH1}$ | ↑ | ↑ | ↑↑ | ≈ | ↓↓ | ↑ | |
| | $V_{TH2}$ | ↓ | ↓ | ↓↓ | ≈ | ↑↑ | ↓ | |
| | $g_{m1}$ | ↑↑ | ↑↑ | ↑↑ | ↓ | ↓↓ | ↑↑ | |
| | $g_{m2}$ | ≈ | ≈ | ≈ | ≈ | ≈ | ≈ | |
| | $C_{vol,1}$ | ≈ | ≈ | ≈ | ↓↓ | ≈ | ≈ | |
| | $C_{vol,2}$ | ≈ | ≈ | ≈ | ↓ | ≈ | ≈ | |
| Biochemical | $c_{ion,1}$ | ↓↓ | ↓ | ↓↓ | ↑ | ↑ | ≈ | |
| | $c_{ion,2}$ | ↑ | ≈ | ↑ | ↑ | ↓ | ↑ | |
| | $\|Z_M\|$ | ≈ | ≈ | ≈ | ≈ | ≈ | ≈ | ↓ |
| Geometrical | $W_1$ | ↑↑ | ↑↑ | ↓ | ↓ | ↓↓ | ↑↑ | |
| | $W_2$ | ↓ | ↓ | ≈ | ≈ | ≈ | ≈ | |
| | $L_1$ | ↓↓ | ↓↓ | ↑ | ↑ | ↑↑ | ↓↓ | |
| | $L_2$ | ≈ | ↓ | ≈ | ≈ | ≈ | ≈ | |
| | $t_1$ | ↑↑ | ↑↑ | ↑↑ | ↓ | ↓↓ | ↑↑ | |
| | $t_2$ | ≈ | ≈ | ≈ | ≈ | ≈ | ≈ | |
| Device | $R_1$ | ≈ | ↓↓ | ↓ | ↓ | ↑ | ↑ | |
| | $R_2$ | ↑↑ | ↑↑ | ↑↑ | ≈ | ↓↓ | ↑↑ | |
| | $V_{IN}$ | ≈ | ≈ | ↑↑ | ↑↑ | ↑↑ | ≈ | |

The symbols represent the relative variation of the figure of merit related to an increase in the corresponding parameter. Parameters are clustered as OMIEC and OECT material parameters, OAN biochemical parameters, OECT geometrical parameters, and OAN device parameters. $A_{Vspike}$ is the voltage amplitude of the spike, $A_{Ispike}$ is the current amplitude of the spike, $V_U$ is the voltage of the crossing point U between the OEND characteristic and the load line, $f_{spike} \times A_{Vspike}$ is the product of the spiking frequency and the voltage amplitude of the spike, $P_{OAN}$ is the power consumed by the OAN, $E_{spike}$ is the Energy per spike and $\tau_M$ is the time constant of the oscillation decay when an intact cellular barrier is interfaced with the OAN.

**Table 2 | Overview of artificial spiking neurons**

| Material | Phenomena | Neuron topology | Components (#) | Neuron in-liquid ion-sensing and bio-interfacing | Spiking frequency (Hz) | Power consumption (μW) | Ref. |
|---|---|---|---|---|---|---|---|
| Silicon | Electronic | Multi-component IC | Multi-T | No | ~1–270 | $75 \times 10^{-3}$ | 65 |
| Silicon | Electronic | SOI MOSFET | 1T | No | ~1–800 | $\sim 22 \times 10^{-3}$ | 1 |
| Silicon | Electronic | Bi-stable resistor | 1T, 1C | No | ~1–350 | NA | 50 |
| Metal-oxide | Mott MIT | Diode-like, NDR | 2D, 2R, 2C-3C, 2 $V_{int}$ | No | $\sim 5 \times 10^3 – 50 \times 10^3$ | 2 | 66 |
| Metal-oxide | Mott MIT | Diode-like, NDR | 1D, 1R, 1C | No | $\sim 30 \times 10^3 – 70 \times 10^3$ | NA | 16 |
| Metal-oxide | Mott MIT | Diode-like, NDR | 1D | No | $\sim 2 \times 10^6 – 9 \times 10^6$ | NA | 67 |
| Organic | Electronic | Diode-like, NDR | 1D | No | $\sim 2 \times 10^6 – 9 \times 10^6$ | NA | 68 |
| Organic | Iono-electronic | Transistor, inverter | 4T, 1R, 1C | Yes | 80 | 60 | 35 |
| Organic | Iono-electronic | Transistor-like, NDR | 2T, 2R, 1C | Yes | 5–55 | 24 | 34 |

Artificial spiking neurons providing biophysical realism capabilities (viz. emulation of real neuron spiking behavior). Spiking neurons fabricated with various materials, underling the key phenomena, the neuron circuit topology, and the number of components. The comparison includes the ability of neurons to perform in-liquid ion-sensing and bio-interfacing, with biochemical-dependent spiking activities. Finally, spiking frequency, power consumption, and biophysical realism, viz. the ability to emulate realistically the electrical response of biological neurons, are provided.
*NDR* negative differential resistance, *IC* integrated circuit, *SOI* silicon on insulator, *MOSFET* metal-oxide-semiconductor field-effect transistor, *T* transistor, *C* capacitor, *R* resistor, *D* diode, *NA* not available.

concentration of the electrolytic medium can be precisely designed to reproduce the behavior of real neurons. A biohybrid OAN obtained by integrating a cellular barrier has been investigated by means of transient non-linear simulations. The impact of the cellular barrier on the OAN characteristics has been quantified. The analytical expressions describing key aspects of neuronal behavior agree with both measurements and numerical simulations and link the OAN spiking features and its materials/physical parameters, thus bringing closer the domains of artificial neurons and neuroscience. This work provides streamlined and transferable guidelines for the design, development, engineering, and optimization of organic artificial neurons, pushing forward a community-aware and community-wide development of the next-generation OAN-based neural networks, neuromorphic electronics, and intelligent bioelectronics.

## Methods
### Device/OAN fabrication
Standard microscope glass slides (75 mm × 25 mm) were cleaned in a sonicated bath, first in a soap solution (Micro-90 (Sigma–Aldrich)) and then in a 1:1 (vol/vol) solvent mixture of acetone and iso-propanol. Source and drain electrodes were made with photolithographically patterned gold (with positive Microposit S1813 photoresist (DOW)) on the cleaned glass slides. A chromium layer was used to improve the adhesion of gold. Each glass slide contains two OECTs $T_1$ and $T_2$, with connections to their respective contacts. The channel dimensions of $T_1$ and $T_2$ are $W_1 \times L_1 = 50\,\mu m \times 20\,\mu m$ and $W_2 \times L_2 = 50\,\mu m \times 10\,\mu m$, respectively. A separate Ag/AgCl electrodes was used for separately gating the OECTs via aqueous electrolytes. Two layers of parylene C (SCS Coatings) were deposited. Soap

(Micro-90 soap solution, 1% vol/vol in deionized water) was used for separation between the parylene C layers to enable the peel-off of the upper parylene C layer. The lower parylene C layer insulates the gold electrodes. Silane A-174 (γ-methacryloxypropyl trimethoxysilane) from Sigma–Aldrich was added to the lower parylene C layer to enhance adhesion. The channel dimensions of $T_1$ and $T_2$ were defined in the second photolithography step through the positive photoresist AZ 9260 MicroChemicals (Cipec Spécialités). Reactive ion etching ($O_2/CF_4$ plasma, 160 W for 16 min with $O_2$ flow rate of 50 s.c.c.m. and $CHF_3$ flow rate of 5 s.c.c.m.) was used to define the channels of $T_1$ and $T_2$ throughout the photoresist mask. $T_1$ channel is made with the organic mixed ionic–electronic conductor polymer PEDOT:PSS (Clevios PH 1000) mixed with 5.0 wt% ethylene glycol, 0.1 wt% dodecyl benzene sulfonic acid, and 1.0 wt% (3-glycidyloxypropyl)trimethoxysilane. Spin coating was used to produce a film in two steps at 1500 rpm and 650 rpm for 1 min and annealed at 120 °C for 1 min in between. T2 channel is made with the semiconducting polymer p(g2T-TT), synthesized according to another work[63]. Here p(g2T-TT) was dissolved in chloroform (3 mg ml$^{-1}$) inside an N2-filled glovebox and spin-coated in ambient conditions at 1000 rpm for 1 min resulting in a thickness of 40 nm. The devices were baked at 60 °C for 1 min. The sacrificial upper parylene C layer was peeled off to confine the polymer to the inside of the channel regions. The devices were subsequently baked at 140 °C for 1 h. Excess soap was rinsed off with deionized water. A diagram of the device fabrication process is provided in Supplementary Fig. 33.

## Numerical simulations

The OECT model reported in Supplementary Note 1 was implemented in a Verilog-A module and used for OAN simulations in the electronic design automation software Advanced Design System (ADS) 2023 Update 1 by Keysight Technologies. The file containing this model was included in a "Verilog" folder inside the ADS 2023 project folder, then a symbol was created and linked to the symbol. The symbol must have the same input/output ports as defined in the Verilog-A module. The user must define the same internal parameters with corresponding names (case sensitive). The symbol created can then be used in any valid circuit topology. The model parameters obtained by reproducing the measured transfer characteristics of $T_1$ and $T_2$ displayed in Supplementary Fig. 1 read: $W_1 = 50\,\mu m$, $L_1 = 20\,\mu m$, $g_{m1} = 1.5 \times 10^{-3}\,S\,V^{-1}$, $V_{TH1} = 0.477\,V$, $\gamma_1 = 2$, $W_2 = 50\,\mu m$, $L_2 = 10\,\mu m$, $g_{m2} = 3.4 \times 10^{-3}\,S\,V^{-1}$, $V_{TH2} = -0.23\,V$, $\gamma_2 = 2.18$. Numerical simulations are performed in DC, AC, transient, and frequency mode. Data analysis of the simulation results was performed by using the software MATLAB 2023a from Mathworks.

## Data availability

The data that support the findings of this study are available from the corresponding authors on request. Source data of the figures in the main paper are provided at the following link: https://doi.org/10.6084/m9.figshare.25968331.

## Code availability

The code used to analyze the simulations obtained is available at GitHub (https://github.com/piethelemon/UnravellingArtificialNeuronsRepo.git)[64]. All code was developed with MATLAB 2023a from Mathworks.

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

## Acknowledgements

This work was financially supported by Ministero dell'Università e della Ricerca (MUR) – project REACH-XY, MUR and Ministero delle Politiche Agricole Alimenatri e Forestali (MIPAAF) - project 1LIVEXYLELLA, Ministero dello Sviluppo Economico (MISE) now Ministero delle Imprese e del Made in Italy (MIMI) - project SMARTCAP. The authors also acknowledge funding from the Carl-Zeiss-Stiftung, via the Emergent AI Center of Johannes Gutenberg University, Mainz, Germany.

## Author contributions

Conceptualization (P.G., and F.T); Methodology (P.B., I.M., P.G., and F.T.); Investigation (P.B., J.P.T., P.G., and F.T.); Visualization (P.B, P.G, and F.T.); Funding acquisition (P.W.M.B., P.G., and F.T.); Project administration (P.G., and F.T.); Supervision (Z.M.K-V., P.G., and F.T.); Writing – original draft (P.G., and F.T.); Writing – review & editing (P.B., P.G., and F.T).

## Funding

## Competing interests

The authors declare no competing interests.
