## [Peer Review File · Nature Communications]

Unravelling the operation of organic artificial neurons for neuromorphic bioelectronicsREVIEWER COMMENTS

Reviewer #1 (Remarks to the Author):

In this manuscript, Belleri and colleagues characterize the operation regimes of OECT-based organic artificial neurons, provide a simulation model, characterize the influence of device parameters on firing rate and energy of these neurons. Lastly, they utilize the device to convert ion concentration into firing rate. Overall, it is a technical paper with detailed description of the device operation regimes however the limited demonstration of its capacity in bio-interfacing applications significantly reduces its accessibility to broader scientific audience.

The manuscript's introduction emphasizes the significance of energy efficiency in neuromorphic circuits. However, it lacks a direct comparison between the energy consumption of these circuits and that of classical circuits or Si-based spiking circuits, especially concerning biological applications. Additionally, the majority of input signals mentioned exceed 1 mA, which is considerably higher than typical biopotentials. The authors should elaborate on the strategies employed to address such a substantial mismatch.

What is the significance of voltage vs current operation-mode of these devices? Is there any unique feature that makes one desired for a targeted application?

It would be helpful if authors add arrows indicating the direction of the sweeps as colors do to not represent directionality.

The reasoning behind the hysteresis of the device is unclear, furthermore I was not able to find any description of how it was implemented into their model.

If I understood correctly, the firing profiles of the neurons are achieved using a ramp input (please add this information to the figure legend). Does the hysteretic profile of the device give rise to the asymmetrical profile of the spikes?

In general, the arrangement of figure 1 and 2 makes the manuscript very confusing. In fact, in the text, even authors had return to figure 1's explanation after completion of figure 2

Given the absence of any biological experiments and the manuscript focus on electrical characterization of these devices, it would be better to focus on power (W) instead of energy to provide a meaningful comparison with other technologies. It will also help to include the duration of the spike itself.

Which device parameter was utilized to adjust the level of excitability? It would be beneficial to include a plot displaying a linear profile of the device parameter against the threshold of excitability. This plot can serve as evidence regarding the degree of tunability.

Defining ion concentration measurement as a "bio-interfacing" application is challenging to justify. In this context, the majority of the circuitry doesn't need to be in the liquid. Moreover, comparable or superior sensitivity can be attained using a straightforward OECT or other transistor architectures. I suggest the authors undertake an experiment that (1) underscores the distinctive capabilities of neuromorphic circuitry in contrast to conventional approaches and (2) underscores the advantages of employing such circuits in conjunction with OECTs

and organic materials.

Reviewer #2 (Remarks to the Author):

The work by Belleri et al. reports on the derivation of the electrostatic equation of OEND devices and the electrical modeling of artificial neurons based on OEND element. The work proposes an extension of the previous paper in Nature electronics, which was the foundation for OEND-based artificial organic neurons. The work represents an important and interesting in-depth analysis of the OEND and OAN device operation for further development of organic neuromorphic circuits. The paper is well written and the work is robust. The main limitation of the work to be published in Nature Communication is the lack of a new finding or new direction offered by the presented results. The paper will be of great interest for people in the community of organic neuromorphic devices.

p9L270-293: it would be interesting to bring the analysis of device parameter done to the material level (i.e. how to relate V_{th} , G_m , C with the different ionic-electronic materials that are known or how to influence the design of new materials to improve OEND and OAN)

p11L314-316: it is not really the electrolyte that is at a given potential, but rather a reference electrode. It is not straightforward to mention that a few mV potential from an external electrode is equivalent to biopotentials. This is strongly related to the impedance of the voltage source

p12L345: not clear what is "range of spatiotemporal scale"

fig.6: is it possible to get the same effect but starting from silent region potential and adding noise to get into the spiking region?

fig.7: The analogy in between extra/intra cellular ionic concentration is difficult to get since there is no intra-cellular domain in the OAN (or what is physically the intracellular domain needs to be precised).

p16L452: is the load line dependent on ionic concentration in the electrolyte? The effective electrolyte resistance is changed, right?

section biointerfacing: The electrical model seems oversimplified. Either the gate is also a capacitor and the question is which capacitor between C_m and C_{gate} see the largest potential, or potential at point B depends on the R_{seal} resistance that is not represented in this schematic (R_{seal} being the resistance between B and ground, which depends on the cell contact, see work from Fromherz for instance). The resulting electrical model analysis is questionable.

Reviewer #3 (Remarks to the Author):

In this manuscript, the authors use numerical simulation tools to model the operation mechanism of organic electrochemical artificial neurons proposed in their previous work (Nature Electronics 2022, 5(11): 774-783.), the results demonstrate a good agreement between the measurements and numerical simulations, providing clear guidelines for the design, development, and optimization of organic artificial neurons constructed by organic electrochemical nonlinear device. This is excellent work and should be published after addressing and/or commenting these points:

1. Since the main experiments in this work closely resemble those conducted in the previous study and considering that the previous work demonstrated the possibility of OAN displaying ion-specific oscillatory activities by incorporating ionophore-based selective membranes, would be possible to model ion-selective OAN? It should be more similar to the information-selective processing of biological carriers.

2. For Table 1, the majority of the impact of parameters on OAN performance is summarized based on experiments, thus more details regarding the estimation of OAN performance based on device geometrical parameters (W, L and thickness) should be included.
3. How extendable is this model to other OMIEC?
4. In the bio-interfacing testing, what is the effect of adding toxin compounds to the electrolyte medium on device performance? Testing results from a control group without cells should be provided.
5. A diagram of detailed device fabrication process and the details of bio-interfacing testing should be provided to enhance clarity for readers.

Response to the Reviewers

We thank the Reviewers for their valuable comments and suggestions, that we found to be extremely helpful in enhancing the quality, and thus the impact, of our manuscript. We thoroughly addressed all the various comments and suggestions. Herewith we provide the point-by-point responses to all of the concerns raised by the Reviewers. In each case we described the changes have been made to the manuscript. All the new results, analysis and changes are highlighted in the revised manuscript. The changes in the revised manuscript are highlighted and a tag indicating the response to Reviewers is also added (e.g. RxQy indicates the response to question x of Reviewer y).

Reviewer #1 (Remarks to the Author):

In this manuscript, Belleri and colleagues characterize the operation regimes of OECT-based organic artificial neurons, provide a simulation model, characterize the influence of device parameters on firing rate and energy of these neurons. Lastly, they utilize the device to convert ion concentration into firing rate. Overall, it is a technical paper with detailed description of the device operation regimes however the limited demonstration its capacity in bio-interfacing applications significantly reduces its accessibility to broader scientific audience.

[R1] Authors' response: We are glad that the Reviewer found our paper to be of interest. Recognizing the significance of demonstrating the broader applicability of our research, we have undertaken comprehensive revisions to the manuscript based on the Reviewer's comments. This includes providing a more detailed overview of the bio-interfacing applications of our OECT-based artificial neurons and addressing the valuable suggestions provided. Below, we provide point-by-point responses, highlighting the changes made in the revised manuscript.

The manuscript's introduction emphasizes the significance of energy efficiency in neuromorphic circuits. However, it lacks a direct comparison between the energy consumption of these circuits and that of classical circuits or Si-based spiking circuits, especially concerning biological applications. Additionally, the majority of input signals mentioned exceed 1 mA, which is considerably higher than typical biopotentials. The authors should elaborate on the strategies employed to address such a substantial mismatch.

[R1Q1] Authors' response: We thank the Reviewer for his/her comment. We agree that in the introduction we emphasized the significance of the energy efficiency in neuromorphic circuits however our work focuses on the wetware operation of organic artificial neurons (OANs). In contrast to Si-based spiking circuits, OANs are based on soft organic mixed ionic electronic conductors (OMIECs) and their associated devices, viz. organic electrochemical transistors (OECTs). OANs possess the remarkable ability to realistically mimic biological phenomena by responding to key biological information carriers such as alkaline ions in electrolytes, noise in the electrolyte, and biological conditions. Therefore, OANs perfectly complement the broad range of features already demonstrated by Si-based spiking circuits, offering opportunities for

both hybrid interfacing between these technologies and new developments in neuromorphic bioelectronics. In our work we unravel the operation of OANs by combining experiments and simulations. After the analysis of the fundamentals governing the OAN operation, we investigate, reproduce, rationalize, and design the wide biorealistic repertoire of organic electrochemical artificial neurons including their firing properties, neuronal excitability to electric, noise, ionic, and biological stimuli.

According with the Reviewer’s comment, we revised the manuscript including a direct comparison between these circuits and a broad range of spiking circuits based on Silicon, Ferroelectric, Ferromagnetic, 2D material, metal-oxide, and organic technologies. Supplementary Table 1 shows a comparison between the various technologies. The analysis comprises power consumption as well as other key parameters including capability of wetware operation in liquid environment, capability of in-situ ion sensing with spiking activity triggered and modulated by ions, capability of in-situ biological interfacing, spiking frequency, and reproduction of biophysical functions. The comparison shows that OANs based on iono-electronic devices with NDR operation are the sole technology combining the capabilities of wetware operation, in-situ ion sensing, and in-situ biological interfacing. Therefore, the class of OANs perfectly complement the broad range of features already demonstrated by solid-state spiking circuits, offering opportunities for both hybrid interfacing between these technologies and new developments in neuromorphic bioelectronics. (see pages 2 and 3, Supplementary Table 1)

Supplementary Table 1 | Comparison between state-of-art spiking circuits. Spiking circuits fabricated with various technologies: silico, ferroelectric, ferromagnetic, 2D materials, hybrid, meat-oxide, and organic. Abbreviations: Mott metal-insulator transition (Mott MIT), negative differential resistance (NDR), integrated circuit (IC), ring oscillator (RO), silicon on insulator (SOI), metal oxide semiconductor field effect transistor (MOSFET), transistor (T) capacitor (C), resistor (R), diode (D), not reported (NR).

Material	Phenomena	Neuron topology	Components (#)	Wetware operation	In-situ ion sensing	In-situ biological interfacing	Spiking frequency (Hz)	Power consumption (μ W)	Biophysical realism	Ref
Silicon	Electronic	Multi-component IC	Multi-T	No	No	No	\sim 1-270	$75 \cdot 10^{-3}$	Yes	1
Silicon	Electronic	Multi-component IC	Multi-T	No	No	No	\sim 200-900	$18 \cdot 10^{-3}$	Partially	2
Silicon	Electronic	SOI MOSFET	1T	No	No	No	\sim 1-800	$\sim 22 \cdot 10^{-3}$	Yes	3
Silicon	Electronic	Bi-stable resistor	1T, 1C	No	No	No	\sim 1-350	NA	Yes	4
Ferroelectric	Polarization	Transistor-like switching	1T	No	No	No	NA	NA	No	5

Ferromagnetic	Spintronic	Diode-like	1D	No	No	No	250·10 ⁶ 400·10 ⁶	1	No	⁶
2D materials	Filament-based	Diode-like, switching	1D	No	No	No	NA	NR	No	⁷
Multi-component	Photonic	Multi-component	Multi-component	No	No	No	1·10 ³ 8.0·10 ³	NR	Partially	⁸
Hybrid	Mott MIT	Diode-like, NDR	1D, 1R, 1C	No	No	No	0.7·10 ⁶ 2.7·10 ⁶	NR	Partially	⁹
Metal-oxide	Mott MIT	Diode-like, NDR	2D, 2R, 2C-3C, 2 V _{int}	No	No	No	~5·10 ³ 50·10 ³	2	Yes	¹⁰
Metal-oxide	Mott MIT	Diode-like, NDR	1D, 1R, 1C	No	No	No	0.4·10 ⁶ 0.9·10 ⁶	28	Partially	¹¹
Metal-oxide	Vacancy-based	Diode-like	1D	No	No	No	NA	35	Partially	¹²
Metal-oxide	Mott MIT	Diode-like, NDR	2D, 3R, 3C	No	No	No	~8·10 ³ 28·10 ³	NR	Partially	¹³
Metal-oxide	Mott MIT	Diode-like, NDR	1D, 1R, 1C	No	No	No	~30·10 ³ 70·10 ³	NR	Yes	¹⁴
Metal-oxide	Mott MIT	Diode-like, NDR	1D	No	No	No	~2·10 ⁶ 9·10 ⁶	NR	Yes	¹⁵
Organic	Electronic	Diode-like, NDR	1D	No	No	No	~2·10 ⁶ 9·10 ⁶	NR	Yes	¹⁶
Organic	Electronic	Inverter-like, RO	9T, 2C, 1R	No	No	No	0.4-5	40	Partially	¹⁷
Organic	Iono-electronic	Inverter-like	5T, 2C	Yes	Yes	No	0.06-0.25	15	Partially	¹⁸
Organic	Iono-electronic	Transistor-like, NDR	2T, 2R, 1C	Yes	Yes	Yes	5-55	24	Yes	¹⁹

Additionally, the majority of input signals mentioned exceed 1 mA, which is considerably higher than typical biopotentials. The authors should elaborate on the strategies employed to address such a substantial mismatch.

[R1Q1b] We agree with the Reviewer that typical biopotentials (measured extracellularly) are lower than 1mV. According with the Reviewer’s comment we revised the manuscript demonstrating phase-locked bursting with excitation signals in the range of microvolts. More in detail, the degree of excitability can be adjusted to the desired level by appropriately configuring the OAN parameters. To explore this important aspect, we conducted numerical simulations, systematically varying the material and device parameters. To evaluate the impact of each parameter, we varied one parameter at a time while keeping all others constant. Supplementary Fig. 18 depicts the excitation threshold voltage, V_{exc}, as a function of OAN

parameters. Excitability increases with g_{m1} , g_{m2} , C_{V1} , C_{V2} , R_1 and R_2 while decreases with V_{TH1} and V_{TH2} . These findings suggest that the level of excitability can be finely tuned, ranging from a few microvolts to hundreds of millivolts. This remarkable degree of tunability is achieved by configuring the OAN parameters and allows, for example, phase-locked bursting with excitation input signals as low as a few microvolts. (Fig. 5f).

Based on the Reviewer's comment, we revised the manuscript incorporating this analysis (pages 14 and 15), and including Fig. 5f and Supplementary Fig. 18.

Supplementary Figure 18 | Excitability as a function of OAN parameters. Investigation of the excitability as a function of as a function of OAN parameters of **a** threshold voltage of T_1 , **b** threshold voltage of T_2 , **c** normalized transconductance of T_1 , **d** normalized transconductance of T_2 , **e** volumetric capacitance of T_1 , **f** volumetric capacitance of T_2 , **g** resistance of R_1 , and **h** resistance of R_2 .

Fig. 5f. Spiking current as a function of time at various input excitation signals $V_{exc} = 0\text{-}40 \mu\text{V}$.

What is the significance of voltage vs current operation-mode of these devices? Is there any unique feature that makes one desired for a targeted application?

[R1Q2] Authors' response: We appreciate the feedback from the Reviewer. Voltage vs current operation-mode of the OEND is a typical measurement conducted for characterizing non-linear devices, such as redox-diffusive memristors, Mott memristors, single-transistor latch, and Gaussian heterojunction transistors⁴⁵. Serving as the non-linear core component of the OAN, the OEND exhibits a behavior like other spiking neuron devices based on non-linear elements, displaying an abrupt increase or decrease in either the current-voltage or voltage-current relationship. The current-voltage characteristic of the OEND is crucial for designing and shaping the spiking behavior of the OAN. Specifically, the OEND can be assessed either as $I_{OEND}(V_{OEND})$

where V_{OEND} is the independent (input) variable (Fig. 1b), or as $V_{\text{OEND}}(I_{\text{OEND}})$ where I_{OEND} is the independent variable (Fig. 1c).

When the OEND is operated in voltage mode, V_{OEND} is the independent input variable: V_{OEND} is swept forward and backward and the current I_{OEND} flowing through the OEND is recorded. As displayed in Supplementary Fig. 3, a hysteretic characteristic is obtained under steady-state operation, which is inherently attributed to the non-linear circuit configuration. In this mode of operation, the switching voltages V_{ON} and V_{OFF} can be directly visualized, and as depicted in Supplementary Fig. 3, the amplitude of the OAN voltage oscillations can be calculated as $A_{V_{\text{spike}}} = V_{\text{ON}} - V_{\text{OFF}}$. This relevant information can be obtained from the OEND characteristic assessed in voltage mode $I_{\text{OEND}}(V_{\text{OEND}})$. The voltages V_{ON} and V_{OFF} are also related to the spiking frequency f_{spike} (Eq. 5), which is a key parameter of the OAN. Finally, we note that the operation of the OAN necessitates that $V_{\text{ON}} > 0 \text{ V}$, $V_{\text{OFF}} > 0 \text{ V}$, and $V_{\text{ON}} > V_{\text{OFF}}$. Equations (1) and (2) demonstrate that these conditions are satisfied when $V_{\text{TH1}} > 0 \text{ V}$, $V_{\text{TH2}} < 0 \text{ V}$, and $V_{\text{TH1}} > |V_{\text{TH2}}|$. Consequently, T_1 has to be a depletion-mode OECT, while T_2 has to be an accumulation-mode OECT. Therefore, accessing the OEND in voltage mode is very relevant to obtain the oscillation voltages, and efficiently design the spiking amplitude and frequency of the OAN.

When the OEND is operated in current mode, I_{OEND} is the independent input variable: I_{OEND} is swept forward and backward and the voltage V_{OEND} is recorded. As displayed in Fig. 3d (blue full line) the $V_{\text{OEND}}(I_{\text{OEND}})$ characteristic is non-hysteretic and shows S-shape negative differential resistance (NDR, blue line between points U_{ON} and U_{OFF} in Fig. Fig. 3d). OAN spiking behavior is obtained when the load-line of the biasing network ($V_{\text{IN}}\text{-}R_{\text{L}}\text{-}C_{\text{L}}$) crosses the OEND characteristic in the NDR region. OAN spiking activity is inhibited when a stimulus causes the crossing point to shift above U_{OFF} . Conversely, spiking activity in the OAN is triggered when a stimulus causes the crossing point to shift from below to above U_{ON} , and the distance between the initial position of the crossing point and U_{L} defines the OAN excitation threshold. Therefore, accessing the OEND in current mode is very relevant to design the OAN operating point, as well as the excitability and inhibition thresholds. Notably, we observe that in our OAN, the crossing point can be shifted by applying stimuli to the liquid electrolyte, which acts as the medium for gate-channel coupling in OECT devices. We demonstrate that injecting noise into the electrolyte, varying ion concentrations, and altering the status of biological membranes lead to changes in the OECT characteristics. These changes are reflected in variations in the S-shaped $V_{\text{OEND}}(I_{\text{OEND}})$ characteristic and, consequently, in the operating point of the OAN. These features make the OAN an ideal component for neuromorphic biosensing and bio-hybrid interfacing.

According to the Reviewer's comment, we revised the manuscript including the explanations and analysis reported above (pages 4, 7, and Fig. 2e)

Supplementary Figure 3 | OEND characteristic and OAN spiking amplitude. Relation between non-linear phenomena and electrochemical oscillations. **a** OEND current-voltage characteristic $I_{OEND}(V_{OEND})$ accessed in voltage mode (symbols and red line): V_{OEND} is the independent input variable and I_{OEND} is the output. OEND current-voltage characteristic $V_{OEND}(I_{OEND})$ accessed in current mode (grey line): I_{OEND} is the independent input variable and V_{OEND} is the output. Symbols are the measurements, red line is calculated with numerical simulations. **b** Measured (symbols) and simulated (full line) spiking voltage (V_{spike}). **c** Measured (symbols) and simulated (full line) spiking current (I_{spike}).

Fig. 5d. OEND (blue full line) and load-line (dashed line) characteristics. When the load-line characteristic crosses the OEND characteristic in the negative differential resistance region (e.g., point

U), its response bifurcates, producing voltage and current oscillations. The OEND negative differential resistance is highlighted by dot-dashed purple line. The spiking region is defined by the upper and lower points $U_L = \{V_{ON}, I_{ON}\}$ and $U_H = \{V_{OFF}, I_{OFF}\}$, respectively. S_1 and S_2 are two points where the OEND characteristic shows a positive resistance.

It would be helpful if authors add arrows indicating the direction of the sweeps as colors do to not represent directionality.

[R1Q3] Authors' response: We agree with the Reviewer that the arrows significantly improve the clarity. According to the Reviewer's comment we revised the manuscript adding arrows indicating the direction of the sweeps in Figs. 1b, 1c, Fig. 2a, 2c, 2d, and 2e.

The reasoning behind the hysteresis of the device is unclear, furthermore I was not able to find any description of how it was implemented into their model.

[R1Q4] Authors' response: We appreciate the feedback from the Reviewer. In the revised manuscript, we elaborate on the rationale behind the hysteresis observed in the OEND device and provide a detailed description of the modelling of the OEND and its hysteretic characteristic. The following explanation is incorporated into the revised manuscript (pages 6-8).

When operating the OEND in voltage mode, V_{OEND} is swept forward and backward, while recording the current I_{OEND} flowing through the OEND. As illustrated in Fig. 2a, under steady-state conditions, a hysteretic $I_{OEND}(V_{OEND})$ characteristic is observed. The OEND exhibits a hysteretic characteristic due to the non-linear current switching from branch $R_1-T_1-R_2$ to branch R_1-T_2 in the forward voltage sweep and from branch R_1-T_2 to $R_1-T_1-R_2$ in the backward voltage sweep.

Specifically, during the forward sweep of V_{OEND} from 0 V to positive voltages lower than V_{ON} , the current I_{OEND} flows through the branch $R_1-T_1-R_2$ (region 1, Fig. 2a). This occurs because T_1 is a depletion-mode p-type OECT ($V_{T1} > 0$ V) and T_2 is an accumulation-mode p-type OECT ($V_{T2} < 0$ V). At small V_{OEND} , T_1 operates in the linear region, resulting in a small channel resistance R_{T1} and consequently a small source-drain voltage $V_{SD1} = R_{T1} \times I_{OEND}$ is obtained. The circuit topology dictates that $V_{SD1} = V_{SG2}$, and hence $V_{SG2} < |V_{TH2}|$ (Fig. 2d). During the forward sweep of V_{OEND} , as long as $V_{OEND} < V_{ON}$ (region 1, Fig. 2a), T_1 remains ON while T_2 remains OFF, causing I_{OEND} to flow through the branch $R_1-T_1-R_2$, resulting in a slope of the $I_{OEND}(V_{OEND})$ characteristic of $dI_{OEND}/dV_{OEND} = 1/(R_1+R_{T1}+R_2)$.

As V_{OEND} increases, the source-drain voltage T_1 (V_{SD1}) also increases. Once $V_{SD1} > |V_{TH2}|$, T_2 turns ON and operates in the saturation region (Fig. 2d). By design, when T_2 is ON, its channel

resistance (R_{T2}) becomes much smaller than R_2 , causing I_{OEND} to predominantly flow through the branch R_1 - T_2 , leading to a sharp increase in current (region 2 in Fig. 2a). This non-linear current enhancement results in a substantial voltage drop across R_1 . Being $V_{R1} = V_{GS1}$ (Fig. 2c), the overdrive voltage of T_1 decreases and eventually $V_{GS1} > V_{TH1}$ causes T_1 to turn OFF (Fig. 2c). The voltage required to turn ON T_2 is referred to as V_{ON} . With further increase in V_{OEND} beyond V_{ON} (region 3 in Fig. 2a), I_{OEND} flows through the branch R_1 - T_2 , with a linear increase characterized by a slope $dI_{OEND}/dV_{OEND} = 1/(R_1+R_{T2})$.

When V_{OEND} is swept back to lower voltages, I_{OEND} linearly decreases, resulting in a decrease in the voltage $V_{GS1} = R_1 \times I_{OEND}$. When $V_{GS1} < V_{T1}$, T_1 switches ON in the linear region. Under this condition, R_{T1} is small, and $V_{SD1} = R_{T1} \times I_{OEND}$ becomes small as well. Given that $V_{SD1} = V_{SG2}$, T_2 turns OFF when $V_{SG2} < |V_{TH2}|$, causing the OEND current to sharply decrease (region 4 in Fig. 2a). The voltage required to turn OFF T_2 is referred to as V_{OFF} .

It is important to note that the hysteresis is observed under steady-state conditions (DC operation) as it is inherently related to the OEND circuit configuration, ensuring that V_{ON} and V_{OFF} occur at different voltages.

When the OEND is operated in current mode, I_{OEND} acts as the independent input variable: I_{OEND} undergoes forward and backward sweeps while the voltage V_{OEND} is recorded. In Fig. 2e the measured data (symbols) and simulated results (line) for the $V_{OEND}(I_{OEND})$ characteristic are presented. The simulations nicely predict the measurements and, unlike the $I_{OEND}(V_{OEND})$ characteristic, the $V_{OEND}(I_{OEND})$ characteristic is non-hysteretic and exhibits S-shape negative differential resistance.

More in detail, at low input currents, I_{OEND} primarily flows through the left branch R_1 - T_1 - R_2 causing V_{OEND} to increase almost linearly with the current (Region 1 in Fig. 2e). When $V_{OEND} = V_{ON}$, T_2 turns ON, I_{OEND} can also flow in the right branch R_1 - T_2 , leading to a lower voltage drop V_{OEND} across the OEND and resulting in a negative differential resistance (Region 2 in Fig. 2e). With further increases in I_{OEND} , the overdrive voltage on T_2 rises ($V_{SG2} = V_{SD1}$), causing the channel resistance R_{T2} to decrease and consequently reducing V_{OEND} until $V_{R1} = I_{OEND} \times R_1$ becomes sufficiently large to deactivate T_1 . Subsequently, current flows solely through the R_1 - T_2 branch, causing V_{OEND} to monotonically increase with I_{OEND} and restoring positive resistance (Region 3 in Fig. 2e). The non-linear partitioning of current between the two branches of the OEND explains the S-shaped non-linear characteristic.

Fig. 2 | OEND operation. **a** OEND electrical characteristic accessed in voltage mode. A voltage ramp (V_{OEND}) is applied forward (red line) and backward (blue line), and the current I_{OEND} is measured. The four operating regions and the relevant circuit components in each region of operation are highlighted. **b** OEND circuit showing the internal voltages. **c** Gate-source voltage of transistor T_1 (V_{GS1}) as a function of V_{OEND} . T_1 threshold voltage (V_{TH1}) is displayed. If $V_{GS1} \geq V_{TH1}$, T_1 is turned OFF while if $V_{GS1} < V_{TH1}$, T_1 is operated in linear regime. Forward sweep (red line) and backward sweep (dashed blue line). The numbers refer to the four operating regions of the OEND, as highlighted in panel a. **d** Source-gate voltage of transistor T_2 (V_{SG2}) as a function of V_{OEND} . If $V_{SG2} \leq |V_{TH2}|$, T_2 is in the OFF state and, if $V_{SG2} > |V_{TH2}|$ results that T_2 is in saturation regime. **e** OEND characteristic accessed in current mode calculated with numerical simulations (line) and measured (symbols). A current ramp (I_{OEND}) is applied forward and backward, and the voltage V_{OEND} is measured. Forward and backward voltages are overlapped. The points $U_{ON} = \{V_{ON}, I_{ON}\}$ and $U_{OFF} = \{V_{OFF}, I_{OFF}\}$ define the negative resistance region (NRD).

If I understood correctly, the firing profiles of the neurons are achieved using a ramp input (please add this information to the figure legend). Does the hysteretic profile of the device give rise to the asymmetrical profile of the spikes?

[R1Q5] Authors' response: We thank the Reviewer for pointing out this relevant point related to the fundamentals of OAN operation. A ramp input is used to measure the current-voltage characteristics of the OEND. We added this information to the figure legend (Fig. 2) and in the main text (page 8). We agree with the Reviewer that the hysteretic profile of the OEND leads to the asymmetrical profile of the spikes. In response to the Reviewer's comment, to better clarify this important point about the OAN operation, we have revised the manuscript including the following analysis (page 9 and Figs. 3a-c).

The spiking activity of the OAN arises from the coupling between the OEND and the biasing network, as depicted in Fig. 3a. When $V_{OEND} < V_{IN}$, the bias current $I_B = (V_{IN} - V_{OEND})/R_L$ charges the capacitor C_L , causing the voltage across the capacitor (V_{spike}) to increase (Fig. 3b, green line). Since the OAN topology gives $V_{spike} = V_{OEND}$, as V_{OEND} increases and the OEND operates in the negative resistance region, the current I_{OEND} decreases, allowing a larger fraction of current I_B ($I_B = I_C - I_{OEND}$) to charge C_L . This further increases V_{OEND} and when $V_{OEND} \geq V_{ON}$ the OEND current significantly increases, reaching the condition $I_{OEND} > I_B$ (Fig. 3c). Subsequently, C_L is discharged and V_{OEND} nonlinearly decreases (Fig. 3b, pink line). When $V_{OEND} \leq V_{OFF}$, I_{OEND} significantly decreases, and when $I_{OEND} < I_B$, C_L is charged again. Therefore, the charging and discharging of the load capacitor depends on the non-linear characteristic of the OEND, input voltage V_{IN} (a DC voltage), and load resistor. The asymmetrical profile of the spikes is a result of the hysteretic profile of the OEND.

Fig. 3 | OAN operation and spiking. **a** OAN circuit highlighting the current partition when V_{spike} increases, and the capacitor is charged. **b** Voltage oscillations (V_{spike}) as a function of time. **c** OAN circuit highlighting the current partition when V_{spike} decreases, and the capacitor is discharged

In general, the arrangement of figure 1 and 2 makes the manuscript very confusing. In fact, in the text, even authors had return to figure 1's explanation after completion of figure 2

[R1Q6] Authors' response: We thank the Reviewer for his/her comment. We revised the text avoiding to return back to figure 1 after completion of figure 2 (page 6).

Given the absent of any biological experiments and the manuscript focus on electrical characterization of these devices, it would better to focus on power (W) instead of energy to provide a meaning full comparison with other technologies. It will also help to include the duration of the spike itself.

[R1Q7] Authors' response: We appreciate the reviewer's suggestion. It has come to our attention that the biological experiments were not adequately presented in the original manuscript. In response to the reviewer's comments, we have revised the manuscript improving the explanation of biological experiments (page 21 and Supplementary Fig. 27).

A biohybrid OAN is obtained by integrating a cellular barrier between the channel and the gate of T_1 . Cell medium is used as electrolyte and the cellular barrier separates the electrolyte into two compartments. A schematic of the biohybrid neuron is shown in Supplementary Fig. 27. The OAN spiking current as a function of time and barrier status is displayed in Fig. 8b (top panel: measurements, bottom panel: simulations). Oscillations with a spiking frequency of 23 Hz are obtained without the cellular barrier (blue line). Upon barrier insertion the oscillations are damped, and OAN firing activity is completely suppressed after about 2 seconds (green line, Fig. 8b). Addition of toxin compounds in the cellular medium give rise to the opening of the TJs and barrier disruption eventually results in a restored spiking of the OAN (pink line, Fig. 8b). The excitability of the OAN can be controlled by the barrier status (Supplementary Fig. 31).

Supplementary Figure 27 | Schematic of the biohybrid OAN. **a** Circuit diagram of the biohybrid OAN. **b** Photograph of the cellular membrane integrated with the OAN. **c** Simplified cross-section of T_1 with

the cellular membrane in the trans-well filter. The gate is inserted in the electrolyte of the apical compartment and the channel is in contact with the electrolyte of the basal compartment. Cell medium is used as electrolyte. **d** Schematic of the epithelial cell layer acting as a physical barrier against ion movement between the basal and apical domains. The cells are interconnected by tight junctions (TJs), forming a barrier that restricts ion flow across the membrane. Ion movement occurs through two potential pathways: the transcellular route (through the cells) and the paracellular route (between the cells). Toxins like hydrogen peroxide (H_2O_2) can disrupt the TJs, impairing the paracellular pathway for ion transport. The equivalent circuit of the bio-membrane is represented by a parallel $R_M C_M$ circuit, with R_M denoting the transmembrane resistance and C_M representing the membrane capacitance. Toxins such as H_2O_2 can compromise the integrity of the TJs, leading to increased transmembrane ion permeability and a reduction in both R_M and C_M . In a biohybrid neuron initially biased below the firing threshold, the introduction of H_2O_2 disrupts the TJs of the bio-membrane, consequently initiating firing.

Fig. 8 | In-liquid biointerfacing. b Measurements (top panel) and transient numerical simulations (bottom panel) of the OAN without cellular barrier (blue line), with intact cellular barrier interfaced with the OAN (green line), and with disrupted cellular barrier (red line). Insertion and disruption of the cellular barrier is highlighted with the vertical dashed lines.

Bio-hybrid excitability modulation. Excitability voltage, V_{exc} , as a function of cellular membrane resistance, R_M , which is related to the status of the cellular membrane.

it would better to focus on power (W) instead of energy to provide a meaning full comparison with other technologies.

[R1Q7b] According to the Reviewer's suggestion, we have revised the manuscript including an analysis of the power consumption. Supplementary Fig. 14 illustrates the power consumption of the OAN (P_{OAN}) as a function of the material and device parameters. Specifically, P_{OAN} reduces with an increase in the threshold voltage (V_{TH1} , panel a) and the normalized transconductance (g_{m1} , panel c) of the OECT T_1 . This is because a lower voltage drop occurs across the channel of T_1 when it operates in the linear region. Similarly, increasing R_2 (panel f) reduces the current flowing in the branch R_1 - R_{T1} and R_2 , consequently lowering the power consumption. Conversely, P_{OAN} increases with an increase in V_{TH2} (panel b) since T_2 turns ON at lower voltages. We observed that for P_{OAN} increases with g_{m2} for small values of g_{m2} and then a plateau is obtained (panel d). This is because the current flowing in the branch R_1 - T_2 is limited by R_1 when T_2 is ON and increasing g_{m2} results that $R_{T2} < R_1$. While R_1 limits the current in the R_1 - T_2 branch, the voltage across R_1 contributes to both turning OFF T_1 (reducing P_{OAN}) and turning ON T_2 (increasing P_{OAN}), explaining the slight increase in P_{OAN} with R_1 (panel e). In addition, the power consumption as a function of the geometrical parameters of transistors T_1 and T_2 is shown in Supplementary Fig. 15.

Supplementary Figure 14 | OAN power consumption vs material and device parameters. Investigation of the OAN power consumption as a function of **a** threshold voltage of T_1 , **b** threshold voltage of T_2 , **c** normalized transconductance of T_1 , **d** normalized transconductance of T_2 , **e** volumetric capacitance of T_1 , **f** volumetric capacitance of T_2 , **g** resistance of R_1 , and **h** resistance of R_2 .

Supplementary Figure 15 | OAN power vs geometrical parameters. Investigation of the OAN power consumption as a function of as a function of OMIEC geometrical parameters: **a** width of T_1 , **b** width of T_2 , **c** length of T_1 , **d** length of T_2 , **e** thickness of T_1 , **f** thickness of T_2 .

Which device parameter was utilized to adjust the level of excitability? It would be beneficial to include a plot displaying a linear profile of the device parameter against the threshold of excitability. This plot can serve as evidence regarding the degree of tunability.

[R1Q8] Authors' response: We are grateful to the Reviewer for this important suggestion. The level of excitability depends on the location of the OAN operating point, which is given by the crossing point between the S-shaped characteristic $V_{\text{OEND}}(I_{\text{ONED}})$ of the OEND and the load-line of the biasing circuits (e.g. U in Fig. 3d). OAN is silent when the operating point is located below the inflection point U_{ON} , while spiking activity is obtained when the operating point is located in the negative differential resistance region, viz. between inflection point U_{ON} and U_{OFF} in Fig. 3d. The OAN excitability is related to the distance from the operating point U and U_{ON} when the operating point is below U_{ON} or to the distance from the operating point and U_{OFF} when

the operating point is above U_{OFF} . The degree of excitability can be adjusted to the desired level by appropriately configuring the OAN parameters, which modify the operating point. To explore this important aspect, we conducted numerical simulations, systematically varying the material and device parameters. To evaluate the impact of each parameter, we adjusted one parameter at a time while keeping all others constant. Supplementary Fig. 18 depicts the excitation threshold voltage, V_{exc} , as a function of variations in OAN parameters. Excitability increases with g_{m1} , g_{m2} , C_{V1} , C_{V2} , R_1 and R_2 while decreases with V_{TH1} and V_{TH2} . The findings suggest that the level of excitability can be finely tuned, ranging from a few microvolts to hundreds of millivolts. This remarkable degree of adjustability is achieved by carefully configuring the OAN parameters allowing, for example, phase-locked bursting with excitation input signals as low as a few microvolts. (Fig. 5f).

According with the Reviewer's suggestion we included the analysis of excitability threshold as a function of the device parameters (page 14, Supplementary Fig. 18, Supplementary Fig. 31)

Supplementary Figure 18 | Excitability as a function of OAN parameters. Investigation of the excitability as a function of as a function of OAN parameters of **a** threshold voltage of T_1 , **b** threshold voltage of T_2 ,

c normalized transconductance of T_1 , **d** normalized transconductance of T_2 , **e** volumetric capacitance of T_1 , **f** volumetric capacitance of T_2 , **g** resistance of R_1 , and **h** resistance of R_2 .

Bio-hybrid excitability modulation. Excitability voltage, V_{exc} , as a function of cellular membrane resistance, R_M , which is related to the status of the cellular membrane.

Defining ion concentration measurement as a "bio-interfacing" application is challenging to justify. In this context, the majority of the circuitry doesn't need to be in the liquid. Moreover, comparable or superior sensitivity can be attained using a straightforward OECT or other transistor architectures. I suggest the authors undertake an experiment that (1) underscores the distinctive capabilities of neuromorphic circuitry in contrast to conventional approaches and (2) underscores the advantages of employing such circuits in conjunction with OECTs and organic materials.

[R1Q9] Authors' response: We thank the Reviewer for his/her valuable comments and suggestions. According to the Reviewer's suggestion we revised the manuscript underscoring the distinct capabilities of neuromorphic circuitry in contrast to conventional approaches as well as the advantages of employing such circuits in conjunction with OECTs and organic materials. Specifically, we added panel f in Fig. 7, we added Supplementary Figs. 23 and 24, and on page 18 we added the following analysis.

A key distinctive capability of neuromorphic circuits is their intrinsic robustness against interfering signals. To this aim, noise is intentionally injected into the electrolyte of the OAN to observe its effect on the spiking behavior at different ion concentrations (Supplementary Fig. 23). While the amplitude of the spiking signal is minimally influenced by noise, the information encoded in the spiking frequency remains unchanged. This is evident in Fig. 7e, where the spiking frequency as a function of ion concentration shows nearly identical curves regardless of the presence of noise. To further illustrate this point, we compare our neuromorphic ion-sensing approach with conventional methods based on OECTs.⁵⁴⁻⁵⁶ When noise of same amplitude and spectral characteristics as that used in the OAN case is injected into the electrolyte of the OECT, significant differences emerge as shown in the transfer characteristics

(I_D - V_G) as a function of ion concentration (Supplementary Fig. 24). Unlike the OAN, the OECT drain current is substantially impacted by noise, leading to corruption of information regarding ion concentration. This comparison underscores the intrinsic robustness of neuromorphic approaches, which encode information in the frequency domain, over conventional methods.

Fig. 7 | Neuromorphic ion sensing. **e** Spiking frequency as a function of ion concentration without and with noise.

Supplementary Figure 23 | Neuromorphic ion sensing. Spiking current as a function of the ion concentration ($c = 3 \cdot 10^{-3} \text{ M}$, $9.5 \cdot 10^{-3} \text{ M}$, $30 \cdot 10^{-3} \text{ M}$) without and with noise injected in the electrolyte of the OAN.

Supplementary Figure 24 | OECT-based ion sensing. OECT drain current as function of the ion concentration ($c = 3 \cdot 10^{-3} \text{ M}$, $9.5 \cdot 10^{-3} \text{ M}$, $30 \cdot 10^{-3} \text{ M}$) without and with noise.

We appreciate the insightful comments from the Reviewer, which have helped us in enhancing the clarity, accuracy, and depth of our work. By addressing the Reviewer's comments and suggestions, we have expanded the investigations and analyses, making the work more accessible to a wider scientific audience.

Reviewer #2 (Remarks to the Author):

The work by Belleri et al. reports on the derivation of the electrostatic equation of OEND devices and the electrical modelling of artificial neurons based on OEND element. The work proposes an extension of the previous paper in Nature electronics, which was the foundation for OEND-based artificial organic neurons. The work represents an important and interesting in-depth analysis of the OEND and OAN device operation for further development of organic neuromorphic circuits. The paper is well written and the work is robust. The main limitation of the work to be published in Nature Communication is the lack of a new finding or new direction offered by the presented results. The paper will be of great interest for people in the community of organic neuromorphic devices.

[R2] Authors' response: We sincerely appreciate the Reviewer's thorough evaluation of our manuscript and are pleased to hear your positive remarks regarding the robustness and clarity of our work. We are grateful for your acknowledgment of the importance of our study in the context of organic neuromorphic devices. We also acknowledge the insightful comments regarding the potential limitations of our study, at least in the original form. In response to this, we have carefully addressed all your valuable comments and suggestions. We addressed your feedback constructively. We revised the manuscript and now provide further insights and novel perspectives, thereby enhancing the overall impact of the work. Below we report the point-by-point responses, highlighting the changes in the revised manuscript. Thank you once again for your valuable feedback and consideration of our manuscript.

p9L270-293: it would be interesting to bring the analysis of device parameter done to the material level (i.e. how to relate V_{th} , G_m , C with the different ionic-electronic materials that are known or how to influence the design of new materials to improve OEND and OAN)

[R2Q1] Authors' response: We thank the Reviewer for the insightful comment. We agree that analyzing device parameters at the material level provides valuable insights into the design and optimization of OEND and OAN devices. In our revised manuscript, we have expanded the analysis highlighting the impact of the material parameters on the OAN performance, to provide a broader understanding of device behavior and guidelines for the design of new materials for future advancements in organic neuromorphic electronics.

According to the Reviewer's suggestion, we analyzed the OAN performance considering the volumetric capacitances C_{V1} and C_{V2} . Results are included in the revised manuscript and displayed in Figs. 4e and 4f, and Supplementary Figs. 6, 10e, 10f, 11e, 11f, 14e, and 14f (also reported below for your convenience).

We found that the volumetric capacitances C_{V1} and C_{V2} of the OMIECs used as channel materials of T_1 and T_2 , respectively, are very significant material parameters influencing f_{spike} , with the spiking frequency being maximum when C_{V1} and C_{V2} are minimized. To further explore

this aspect, Supplementary Fig. 8 (also reported below for your convenience) analyzes the OAN spiking frequency while also considering the relationship between C_{V1} , C_{V2} , and load capacitor C_L . We systematically varied both C_{V1} and C_{V2} , and the minimum C_L required for OAN spiking is calculated. Supplementary Fig. 8a reveals that the minimum C_L amounts to 10^{-7} F and is achieved when C_{V2} falls in the range 50 - 100 $F\text{ cm}^{-3}$ and C_{V1} is approximately 50 $F\text{ cm}^{-3}$. Supplementary Fig. 8b highlights that the maximum spiking frequency, $f_{\text{spike}} = 150$ Hz, is attained when both C_{V2} and C_L are minimized. We observe that minimizing C_{V2} leads to a reduction in the capacitance of OECT T_2 , which can also be achieved by adjusting the geometrical parameters.

In Supplementary Figs. 16 and 17, the analysis of the impact of material parameters on the OAN performance is further extended to several state-of-art OMIECs suitable for depletion-mode (transistor T_1) and accumulation mode (transistor T_2) devices. To assess the impact of each material on the OAN performance, we modified one material at a time while keeping all other parameters constant. The analysis confirms that OMIECs with large volumetric capacitance and low mobility (e.g. p(gNDI-g2T) in Supplementary Fig. 16), provides OANs with reduced spiking frequency, large energy per spike and low power consumption. This is further confirmed in Supplementary Fig. 17 where the material with lowest mobility, i.e. p(gBDT-g2T), provides the lower spiking frequency and the maximum power consumption. This also suggests that the mobility of OMIECs used for T_2 is important to control the power consumption of the OAN.

According with the Reviewer's comment, we included this analysis in the revised manuscript (page 12, Figs. 4e, and 4f, Supplementary Figs. 8, 16 and 17).

Fig. 4 | OAN spiking frequency and energy. e volumetric capacitance of T_1 , **f** volumetric capacitance of T_2 .

Supplementary Figure 6 | OEND characteristics as a function of material parameters. The OEND is accessed in current mode, $V_{\text{OEND}}(I_{\text{OEND}})$. **a** OMIEC volumetric capacitance of transistor T_1 , C_{v1} , ranges from 33 F cm^{-3} to 67 F cm^{-3} . **b** OMIEC volumetric capacitance of transistor T_2 , C_{v2} , ranges from 40 F cm^{-3} up to 300 F cm^{-3} . **c** OMIEC electronic mobility of T_1 , μ_1 , ranges from $0.8 \text{ cm}^2 \text{ V}^{-1} \text{ s}^{-1}$ to $1.7 \text{ cm}^2 \text{ V}^{-1} \text{ s}^{-1}$. **d** OMIEC electronic mobility of T_2 , μ_2 , ranges from $0.05 \text{ cm}^2 \text{ V}^{-1} \text{ s}^{-1}$ to $2 \text{ cm}^2 \text{ V}^{-1} \text{ s}^{-1}$. The dashed lines in all panels represent the load-line, aiding in showing changes of the crossing point as the parameter varies, while the arrows indicate the direction of parameter increase.

Supplementary Figure 10 | Amplitude of OAN spiking voltage vs materials and device parameters. Investigation of the voltage spiking amplitude A_{Vspike} as a function of OAN materials and device parameters: **a** threshold voltage of T_1 , **b** threshold voltage of T_2 , **c** normalized transconductance of T_1 , **d** normalized transconductance of T_2 , **e** volumetric capacitance of T_1 , **f** volumetric capacitance of T_2 , **g** resistance of R_1 , and **h** resistance of R_2

Supplementary Figure 11 | Amplitude of OAN spiking current vs materials and device parameters. Investigation of the current spiking amplitude A_{spike} as a function of OAN materials and device parameters: **a** threshold voltage of T_1 , **b** threshold voltage of T_2 , **c** normalized transconductance of T_1 , **d** normalized transconductance of T_2 , **e** volumetric capacitance of T_1 , **f** volumetric capacitance of T_2 , **g** resistance of R_1 , and **h** resistance of R_2 .

Supplementary Figure 14 | OAN power consumption vs material and device parameters. Investigation of the OAN power consumption as a function of **a** threshold voltage of T_1 , **b** threshold voltage of T_2 , **c** normalized transconductance of T_1 , **d** normalized transconductance of T_2 , **e** volumetric capacitance of T_1 , **f** volumetric capacitance of T_2 , **g** resistance of R_1 , and **h** resistance of R_2 . P_{OAN} reduces with an increase in the threshold voltage (V_{TH1} , panel a) and the normalized transconductance (g_{m1} , panel c) of the OECT T_1 . This is because a lower voltage drop occurs across the channel of T_1 when it operates in the linear region. Similarly, increasing R_2 (panel f) reduces the current flowing in the branch R_1 - R_{T1} and R_2 , consequently lowering the power consumption. Conversely, P_{OAN} increases with an increase in V_{TH2} (panel b) since T_2 turns ON at lower voltages. We observed that for P_{OAN} increases with g_{m2} for small values of g_{m2} and then a plateau is obtained (panel d). This is because the current flowing in the branch R_1 - T_2 is limited by R_1 when T_2 is ON and increasing g_{m2} results that $R_{T2} < R_1$. While R_1 limits the current in the R_1 - T_2 branch, the voltage across R_1 contributes to both turning OFF T_1 (reducing P_{OAN}) and turning ON T_2 (increasing P_{OAN}), explaining the slight increase in P_{OAN} with R_1 (panel e).

Supplementary Figure 8 | OAN spiking frequency analysis. Investigation of the spiking frequency, f_{spike} , as a function of the volumetric capacitances C_{v1} and C_{v2} of the OECT T_1 and T_2 , respectively. **a** Minimum load capacitor, C_L , required for OAN spiking for each combination of C_{v1} and C_{v2} . **b** f_{spike} as a function of C_{v1} and C_{v2} with minimum C_L . The gray regions indicate cases of non-spiking OAN.

Supplementary Figure 16 | OAN performance with various OMIEC materials for depletion-mode OECTs (T_1). Investigation of the OAN performance considering various OMIEC materials suitable for the fabrication of depletion-mode transistor T_1 . The fabricated OAN is based on the material PEDOT:PSS for transistor T_1 and p(g2T-TT) for transistor T_2 . The star symbols indicate the materials and the corresponding parameters used in our work. The parameters of our materials are the following. PEDOT:PSS*: volumetric capacitance $C_V = 43 \text{ F cm}^{-3}$, hole mobility $\mu = 1.02 \text{ cm}^2 \text{ V}^{-1} \text{ s}^{-1}$, threshold voltage $V_{\text{TH}} = 0.477 \text{ V}$. p(g2T-TT)*: $C_V = 241 \text{ F cm}^{-3}$, $\mu = 0.13 \text{ cm}^2 \text{ V}^{-1} \text{ s}^{-1}$, and $V_{\text{TH}} = -0.23 \text{ V}$. The performance estimated with the state-of-art materials are calculated by substituting the material parameters of transistor T_1 with those of the material under investigation. The parameters of the materials investigated (triangle symbols) are taken from Refs.²⁷⁻³¹, and are the following. PEDOT:PSS: $C_V = 39 \text{ F cm}^{-3}$, $\mu = 1.9 \text{ cm}^2 \text{ V}^{-1} \text{ s}^{-1}$, $V_{\text{TH}} = 0.4 \text{ V}$. PEDOT:TOS: $C_V = 136 \text{ F cm}^{-3}$, $\mu = 0.93 \text{ cm}^2 \text{ V}^{-1} \text{ s}^{-1}$, and $V_{\text{TH}} = 0.52 \text{ V}$. p(gNDI-g2T): $C_V = 397 \text{ F cm}^{-3}$, $\mu = 1 \cdot 10^{-4} \text{ cm}^2 \text{ V}^{-1} \text{ s}^{-1}$, and $V_{\text{TH}} = 0.35 \text{ V}$.

Supplementary Figure 17 | OAN performance with various OMIEC materials for accumulation mode OECTs (T_2). Investigation of the OAN performance considering various OMIEC materials suitable for the fabrication of accumulation-mode transistor T_2 . The fabricated OAN is based on the material PEDOT:PSS for transistor T_1 and p(g2T-TT) for transistor T_2 . The star symbols indicate the materials and the corresponding parameters used in our work. The parameters of our materials are the following. PEDOT:PSS*: volumetric capacitance $C_V = 43 \text{ F cm}^{-3}$, hole mobility $\mu = 1.02 \text{ cm}^2 \text{ V}^{-1} \text{ s}^{-1}$, threshold voltage $V_{\text{TH}} = 0.477 \text{ V}$. p(g2T-TT)*: $C_V = 241 \text{ F cm}^{-3}$, $\mu = 0.13 \text{ cm}^2 \text{ V}^{-1} \text{ s}^{-1}$, and $V_{\text{TH}} = -0.23 \text{ V}$. The performance estimated with the state-of-art materials are calculated by substituting the material parameters of transistor T_2 with those of the material under investigation. The parameters of the materials investigated (triangle symbols) are taken from Refs.²⁷⁻³¹, and are the following. p(g2T-TT): $C_V = 241 \text{ F cm}^{-3}$, $\mu = 0.94 \text{ cm}^2 \text{ V}^{-1} \text{ s}^{-1}$, and $V_{\text{TH}} = -0.2 \text{ V}$. p(g2T-T): $C_V = 220 \text{ F cm}^{-3}$, $\mu = 0.28 \text{ cm}^2 \text{ V}^{-1} \text{ s}^{-1}$, and $V_{\text{TH}} = 0 \text{ V}$. p(gBDT-g2T): $C_V = 77 \text{ F cm}^{-3}$, $\mu = 0.018 \text{ cm}^2 \text{ V}^{-1} \text{ s}^{-1}$, and $V_{\text{TH}} = -0.55 \text{ V}$.

p11L314-316: it is not really the electrolyte that is at a given potential, but rather a reference electrode. It is not straightforward to mention that a few mV potential from an external electrode is equivalent to biopotentials. This is strongly related to the impedance of the voltage source

[R2Q2] Authors' response: We agree with Reviewer that this claim should be further clarified. Due to the faradaic nature of the gate electrode that is used in our work (Ag/AgCl), the voltage drop at the gate/electrolyte interface is negligible and the applied voltage can be approximated

with the electrolyte potential (Ref.1, Ref.2, Ref.3). Nevertheless, as pointed out by the Reviewer, it is not straightforward to compare qualitatively forced (external) voltages with biopotentials, because the actual voltage that a sensing unit (such as the OAN) experiences is also related to the impedance of a voltage source. For instance, in the case of a source of a biological activity (a cell), the potential would depend on the cell-to-device impedance. Nevertheless, the characterization in a “simulated” biological environment (with an electrolyte, ions and forced external voltages at the gate electrode), represents the ideal case of a cell-to-device interface, which at least shows the upper limit sensitivity in voltage changes (i.e., ideal excitability). More specifically, whenever it is stated that the OAN is responsive to 1 mV of gate potential, this represents an ideal case of negligible cell-to-device impedance. In reality, excitability will be equal or lower due to the finite (and oftentimes not well-controllable) cell-to-device impedance. However, using as the metric of the excitability in ideal “simulated” biological environments is necessary for circuit engineering in a well-defined, model environment.

Revision: We added the explanation above in the revised manuscript (page 21). Furthermore, the non-polarizable gate electrode has been described as a resistor R_G and included in the model (Fig. 8a).

Ref.1 Bernards, D. A. & Malliaras, G. G. Steady-State and Transient Behavior of Organic Electrochemical Transistors. *Adv Funct Mater* **17**, 3538–3544 (2007).

Ref.2 Tarabella, G. *et al.* Effect of the gate electrode on the response of organic electrochemical transistors. *Appl. Phys. Lett.* **97**, 123304 (2010).

Ref.3 Picca, R. A. *et al.* Ultimately Sensitive Organic Bioelectronic Transistor Sensors by Materials and Device Structure Design. *Adv Funct Mater* **30**, 1904513 (2020).

p12L345: not clear what is "range of spatiotemporal scale" fig.6: is it possible to get the same effect but starting from silent region potential and adding noise to get into the spiking region?

[R2Q3] Authors' response: We meant that noise could be change as a function of time and space. This sentence was not essential and according with the Reviewer's comment we removed the sentence to improve the clarity (page 15 of the revised manuscript).

Yes, it is possible to get into the spiking region from the silent region by adding noise. We added this analysis in the revised manuscript: As illustrated in Supplementary Fig. 19, it is possible to design OANs with spiking activity induced by noise: OAN is initially silent (below the excitation threshold) and noise can trigger the spiking activity at various intensities – ranging from a few spikes to nearly tonic firing – by increasing its excitability. This shows the high degree of reconfigurability inherent in the OAN.

We revised the manuscript including this analysis (page 16, Supplementary Fig. 19)

Supplementary Figure 19 | Noise-induced activity. White noise signal is injected into the electrolyte. **a,b** The OAN is silent and because the noise is below the excitability threshold. **c,d** Increasing the excitability few random spikes are obtained in the cases the amplitude of noise is above the excitability threshold. **e,f** Further increasing the excitability nearly tonic firing is obtained.

fig.7: The analogy in between extra/intra cellular ionic concentration is difficult to get since there is no intra-cellular domain in the OAN (or what is physically the intracellular domain needs to be precised).

[R2Q4] Authors' response: We agree with the Reviewer that in contrast with biological cells, there is no intracellular space at the OAN domain. More precisely, in bio-interfacing scenarios

between biological cells and OANs, the common electrolyte represents a kind of shared extracellular space. Any exchange of information and signaling (for instance of ionic species) between the biological and artificial domain happens via this shared extracellular space. Nevertheless, the OAN is responsive to the concentration range of the most important ionic species (Na^+ , K^+ , Ca_2^+) of the extracellular and intracellular space, although the biological intracellular space is not directly accessible from the OAN. We revised the manuscript adding this explanation (pages 17 and 21). Moreover, Supplementary Fig. 27 provides a detailed schematic and more information of the biohybrid neuron.

Supplementary Figure 27 | Schematic of the biohybrid OAN. **a** Circuit diagram of the biohybrid OAN. **b** Photograph of the cellular membrane integrated with the OAN. **c** Simplified cross-section of T1 with the cellular membrane in the trans-well filter. The gate is inserted in the electrolyte of the apical compartment and the channel is in contact with the electrolyte of the basal compartment. Cell medium is used as electrolyte. **d** Schematic of the epithelial cell layer acting as a physical barrier against ion movement between the basal and apical domains. The cells are interconnected by tight junctions (TJs), forming a barrier that restricts ion flow across the membrane. Ion movement occurs through two potential pathways: the transcellular route (through the cells) and the paracellular route (between the cells). Toxins like hydrogen peroxide (H_2O_2) can disrupt the TJs, impairing the paracellular pathway for ion transport. The equivalent circuit of the bio-membrane is represented by a parallel $R_M C_M$ circuit, with R_M denoting the transmembrane resistance and C_M representing the membrane capacitance. Toxins such as H_2O_2 can compromise the integrity of the TJs, leading to increased transmembrane ion permeability and a reduction in both R_M and C_M . In a biohybrid neuron initially biased below the firing threshold, the introduction of H_2O_2 disrupts the TJs of the bio-membrane, consequently initiating firing.

p16L452: is the load line dependent on ionic concentration in the electrolyte? The effective electrolyte resistance is changed, right?

[R2Q5] Authors' response: We agree with the Reviewer that the effective electrolyte resistance changes with the ion concentration. This impacts on transistors T_1 and T_2 . The load line is independent of the ion concentration in the electrolyte because R_L is not affected by the ion concentration. Conversely, the OEND characteristics in the negative differential resistance region are significantly modulated by the ion concentration and the crossing point with the load-line characteristic (dashed line in Fig. 7g) systematically shifts with the ion concentration. We clarify this point in the revised manuscript (page 19).

Fig. 7g. OEND characteristics modulated by the ion concentration (solid lines). Load line (dashed dark line) crosses the OEND characteristics in various points inside the spiking region (orange area), depending on ion concentration. At large ion concentration the crossing point $U = \{V_U, I_U\}$ is outside the spiking region and the OAN is silent.

*section biointerfacing: The electrical model seems oversimplified. Either the gate is also a capacitor and the question is which capacitor between C_m and C_{gate} see the largest potential, or potential at point B depends on the R_{seal} resistance that is not represented in this schematic (R_{seal} being the resistance between B and ground, which depends on the cell contact, see work from Fromherz for instance, see *Biophys J.* 2007 Feb 1; 92(3): 1096–1111). The resulting electrical model analysis is questionable.*

[R2Q6] Authors' response: We agree with the Reviewer that the electrical model is a simplification of the reality. First of all, for simplicity, we omit frequency dependent elements such as ion channel conductances. However, there are valid assumptions, along with experimental modifications that strongly represent this simplification. To summarize:

- Because PEDOT:PSS is highly conducting, the impedance spectrum of PEDOT:PSS electrodes is the same with PEDOT:PSS transistors (when connecting Source and Drain in the same potential).[Ref.1]
- We use Ag/AgCl as gate electrode and the voltage drop at the gate/electrolyte interface is negligible and therefore the applied voltage can be approximated with the electrolyte potential.[Ref.1-Ref.4] This means that the gate-to-membrane capacitor can be omitted.

Indeed, Impedance Spectroscopy (IS) in plain PEDOT:PSS electrodes shows an ideal, R_{EL} - C_{PEDOT} in series behavior (where R_{EL} the electrolyte resistance and C_{PEDOT} the PEDOT:PSS electrode capacitance), without any indication of additional gate-to-channel capacitance apart from C_{PEDOT} (see Fig. R1a).[Ref.5-Ref.8] Moreover, the fact that from IS and C_{PEDOT} , the material-related (for PEDOT:PSS) volumetric capacitance is extracted, further proves the validity of this assumption.

- The experimental set-up is slightly different than the one described in Biophys J. 2007 Feb 1; 92(3): 1096–1111. First of all, there is no direct/physical contact between the membrane and the electrode (or the device channel), as the cells are grown on transwell filters and there is a physical/macroscopic separation between the filter and the electrode (<1 cm).[Ref.9-Ref.11] This physically separated gap is filled with the very same electrolyte (cell culture medium) as the gate-to-membrane compartment. In this case (Fig. R1b), the IS spectrum shows a simple typical electrode behavior ($R_{EL}C_{PEDOT}$ in series, with $R_{EL} \sim R_{Ea} + R_{Eb}$) with no obvious signature of the membrane presence ($R_M C_M$ element). In this configuration, we further reduce the area of the transwell filter to $\sim 1/15$ in order to increase the impedance of the membrane ($R_M C_M$). With this area modification, the impedance of the membrane ($R_M C_M$) is much higher than the contribution of background resistance ($R_G + R_{Ea} + R_{Eb}$), and therefore, the background contribution of the other circuit elements is heavily suppressed (see Fig. R1b).

- Under the above mentioned assumptions and conditions, the equivalent circuit of the system can be simplified into that of Figure 8(a) in the revised manuscript.

Fig. R1. (a) Impedance spectroscopy (IS) of a PEDOT:PSS electrode with area $200 \times 200 \mu\text{m}^2$. The electrode shows an ideal impedance spectrum of an $R_{EL}C_{PEDOT}$ in series, with R_{EL} the electrolyte resistance and C_{PEDOT} the PEDOT:PSS electrode capacitance. (b) Is spectrum of a PEDOT:PSS electrode with a transwell filter and a membrane with original area and reduced area. The Impedance Spectrum of the membrane with large area has a typical electrode behaviour (RC in series) with no obvious signature of the membrane presence ($R_M C_M$ element). In this case the background resistance $R_{GE} + R_{EC}$ dominates the spectrum. When the filter area is reduced, the membrane impedance prevails and clearly appears in the spectrum, while the contribution of the background resistance is much lower.

- Ref.1 Koutsouras, D. A. *et al.* An Iontronic Multiplexer Based on Spatiotemporal Dynamics of Multiterminal Organic Electrochemical Transistors. *Adv. Funct. Mater.* **31**, 2011013 (2021).
- Ref.2 Bernardis, D. A. & Malliaras, G. G. Steady-State and Transient Behavior of Organic Electrochemical Transistors. *Adv. Funct. Mater.* **17**, 3538–3544 (2007).
- Ref.3 Tarabella, G. *et al.* Effect of the gate electrode on the response of organic electrochemical transistors. *Appl. Phys. Lett.* **97**, 123304 (2010).
- Ref.4 Picca, R. A. *et al.* Ultimately Sensitive Organic Bioelectronic Transistor Sensors by Materials and Device Structure Design. *Adv. Funct. Mater.* **30**, 1904513 (2020).
- Ref.5 Koutsouras, D. A. *et al.* Probing the Impedance of a Biological Tissue with PEDOT:PSS-Coated Metal Electrodes: Effect of Electrode Size on Sensing Efficiency. *Adv. Funct. Mater.* **23**, 1901215 (2019).
- Ref.6 Rivnay, J. High-performance transistors for bioelectronics through tuning of channel thickness. *Sci. Adv.* **1**, e1400251 (2015).
- Ref.7 Koutsouras, D. A. *et al.* Impedance Spectroscopy of Spin-Cast and Electrochemically Deposited PEDOT:PSS Films on Microfabricated Electrodes with Various Areas. *ChemElectroChem* **4**, 2321-2327 (2017).
- Ref.8 Koutsouras, D. A. Probing the Impedance of a Biological Tissue with PEDOT:PSS-Coated Metal Electrodes: Effect of Electrode Size on Sensing Efficiency. *Adv. Healthcare Mater.* **8**, 1901215 (2019).
- Ref.9 Jimison, L. H. *et al.* Measurement of Barrier Tissue Integrity with an Organic Electrochemical Transistor. *Adv. Mater.* **24**, 5919-5923 (2012).
- Ref.10 Lingstedt, L. V. *et al.* Monitoring of Cell Layer Integrity with a Current-Driven Organic Electrochemical Transistor. *Adv. Healthcare Mater.* **8**, 1900128 (2019).
- Ref.11 Sarkar, T. *et al.* An organic artificial spiking neuron for in situ neuromorphic sensing and biointerfacing. *Nat. Ele.* **5**, 774–783 (2022).

Authors' revision:

According to the explanation provided above, we revised the section “Biointerfacing” on page 21. The model in Fig. 8a has been modified highlighting the resistance of the gate electrode (R_G) and the ionic resistance of the apical (R_{Ea}) and basal electrolyte (R_{Eb}). We highlighted that more complex circuits like in Biophys J. 2007 Feb 1; 92(3): 1096–1111 can be necessary when other experimental set-ups and/or biological systems are used. We revised Supplementary Fig. 28 highlighting the measured impedance without and with the cellular barrier, and the text has been amended as follows:

As detailed in Fig. 8a, the gate electrode is modelled as a resistor, R_G , in series to the ionic resistance of the electrolyte in the apical compartment, R_{Ea} , due to the ion transport from the gate to the cellular barrier in the electrolyte medium of the apical compartment. The cellular barrier is described by a capacitor C_M in parallel to a resistor R_M , and the ionic transport in the electrolyte medium of the basal compartment is modelled by the resistor R_{Ec} . Electrochemical impedance spectroscopy (EIS) measurements in Supplementary Fig. 28 show that the impedance of intact cellular barrier is much larger than the gate and ionic resistance of apical and basal electrolytes, $|Z_M| \gg R_G + R_{Ea} + R_{Eb}$. Focusing on cellular barrier model, C_M accounts

for the ion accumulation at the basal and apical barrier interfaces. C_M is relevant when the TJs are intact since in this condition the ions are not flowing through the barrier. R_M accounts for the ionic transport through the barrier and it is relevant when TJs are disrupted with toxins or transiently open with neuromodulators.⁶³ The barrier parameters are obtained from EIS measurements (Supplementary Fig. 28). It is important to observe that the complexity of the equivalent circuit required may vary depending on the types of cells interfacing with the OAN.⁶⁴

Supplementary Figure 28 | Bio-membrane electrochemical impedance spectroscopy. Measurements (symbols) and model (solid lines) of the biomembrane impedance. Impedance spectrum without the cellular barrier membrane (blue circles) shows a resistor-capacitor (R_B - C_P) circuit, where $R_B = R_G + R_{Ea} + R_{Eb}$, R_G is the gate resistance, R_{Ea} is the apical electrolyte resistance, R_{Eb} is the basal electrolyte resistance and C_P is the capacitance of the OMIEC channel. As a further confirmation, C_P extracted from impedance spectroscopy is normalized to the volume of the PEDOT:PSS, obtaining a volumetric capacitance of 43 F/cm^3 , in full agreement with the state.³²⁻³⁴ The membrane equivalent circuit comprises a resistor R_M in parallel with a capacitor C_M . The cellular membrane parameters obtained from the electrochemical impedance spectroscopy are $R_M = 1.75 \text{ k}\Omega$ and $C_M = 470 \text{ nF}$. Importantly, we note that the membrane resistance R_M is about one orders of magnitude larger than the background resistance R_B .

Reviewer #3 (Remarks to the Author):

In this manuscript, the authors use numerical simulation tools to model the operation mechanism of organic electrochemical artificial neurons proposed in their previous work (Nature Electronics 2022, 5(11): 774-783.), the results demonstrate a good agreement between the measurements and numerical simulations, providing clear guidelines for the design, development, and optimization of organic artificial neurons constructed by organic electrochemical nonlinear device. This is an excellent work and should be published after addressing and/or commenting these points:

[R3] Authors' response: We sincerely appreciate the Reviewer's positive feedback on our manuscript. We are delighted that the Reviewer found our work excellent and recommended the publication after revision. We carefully addressed all the points raised by the Reviewer to further enhance the quality and impact of our work. Below, we provide point-by-point responses, highlighting the changes made in the revised manuscript.

1. Since the main experiments in this work closely resemble those conducted in the previous study and considering that the previous work demonstrated the possibility of OAN displaying ion-specific oscillatory activities by incorporating ionophore-based selective membranes, would be possible to model ion-selective OAN? It should be more similar to the information-selective processing of biological carriers.

[R3Q1] Authors' response: We are grateful to the Reviewer for the relevant suggestion. We revised the manuscript including the following modelling and analysis (page 18, Fig. 7f, Supplementary Figs. 25 and 26, Supplementary Note 1).

Adding another layer of biophysical realism to the OAN response involves integrating selectivity to specific ions, thus obtaining ion-selective excitation and spiking. OANs capable of exhibiting ion-specific oscillatory activities are realized by integrating ionophore-based selective membranes⁵⁷ (ISM) at the interface between the channel and electrolyte of T_1 . The ISM generates a voltage at the membrane/electrolyte interface (V_{ISM}) in response to the specific ion concentration within the electrolyte (Supplementary Note 1). The integration of ISM in the OAN architecture results in a variation of the OECT T_1 threshold voltage V_{TH1} (Supplementary Fig. 25). Being the OAN excitability dependent on V_{TH1} (Supplementary Fig. 18) variation of the target ion concentration results in ion-selective excitability and spiking of the OAN. Fig. 7f shows that a K^+ selective OAN exhibits oscillations when immersed in a KCl electrolyte, with the frequency of oscillations increasing with the concentration of the selected ion type. As a control experiment, Na^+ are used as interfering ions and the OAN is silent (Fig. 7f), demonstrating ion selective spiking activity. Analogously, a Na^+ selective OAN exhibits oscillations when immersed in a NaCl electrolyte and is silent in KCl (Supplementary Fig. 26).

In Supplementary Note 1 we added the model of ion-selective membrane:

When an ion-selective membrane (ISM) is used, the voltage V_{ISM} generated in response to the specific concentration of target ions within the aqueous electrolyte at the analyte/membrane interface can be calculated with the Nernst equation^{24–26} and results:

$$V_{\text{ISM}} = V^0 + s_i \log_{10} \left\{ [M] - \alpha [N] \frac{s_j}{s_i} \right\} \quad (10)$$

where V^0 denotes the formal potential of the electrolyte ion and the work function generated by the connection of materials with different work functions in the overall extended gate architecture, $s_i = \eta_i \frac{k_B T}{z_i e} \ln(10)$, k_B is Boltzmann's constant, T is the temperature, z_i is the valence of the selected ions, e is the elementary charge, η_i is a dimensionless factor between 0 and 1 that accounts for the activity of the selected ions, $[M]$ is the concentration of the selected ions, α is the selectivity coefficient, $s_j = s_i \frac{\eta_j z_i}{\eta_i z_j}$ is the sensitivity of the interfering ions, η_j accounts for the activity of the interfering ions, and $[N]$ is the concentration of interfering ions.

Fig. 7 | Neuromorphic ion sensing. f Ion-selective spiking: spiking frequency as a function of selected K^+ concentration. The OAN is not spiking when the concentration of interfering ions (Na^+) is varied.

Supplementary Figure 25 | Ion selective OECTs. Measured transfer characteristics (symbols) at various **a** potassium (K^+) and **b** sodium (Na^+) concentration. $V_D = -0.1V$. A K^+ -selective ISM is used. Full lines are calculated with the drain-current model in Supplementary Note 1. **c** Threshold voltage variation $\Delta V_{TH} = V_{TH}(c) - V_{TH}(c_{min})$ as a function of $c=K^+$ and $c=Na^+$ concentration. K^+ are the target ions and Na^+ are the interfering ions. Dashed line is calculated with Eq. (10) in Supplementary Note 1. The ISM model parameters extracted from the measurements are the following: $s_i = -52.2 \text{ mV dec}^{-1}$, $s_j = -40 \text{ mV dec}^{-1}$, $\alpha = 0.164$, $V_{K^+}^0 = 0.432 \text{ V}$, and $V_{Na^+}^0 = 0.474 \text{ V}$.

Supplementary Figure 26 | Ion-selective spiking. Spiking frequency as a function of selected Na^+ concentration. The OAN is not spiking when the concentration of interfering ions (K^+) is varied.

2. For Table 1, the majority of the impact of parameters on OAN performance is summarized based on experiments, thus more details regarding the estimation of OAN performance based on device geometrical parameters (W , L and thickness) should be included.

[R3Q2] Authors' response: According with the Reviewer's suggestion we expanded the analysis providing more details regarding the estimation of OAN performance based on device geometrical parameters. To assess the impact of each geometrical parameter on the OAN performance, we systematically varied the OECT geometries, i.e. width (W_1 , W_2), length (L_1 , L_2), thickness (t_1 , t_2). We modified one parameter at a time while keeping all other parameters constant. The parameters are varied accounting for the largest range that ensures OAN oscillation. The corresponding I_{OEND} - V_{OEND} characteristics accessed in current mode are displayed in Supplementary Fig. 7. The relationship between spiking frequency and device geometrical parameters is depicted in Supplementary Fig. 9. It is observed that reducing the dimensions such as W_1 , W_2 , L_2 , t_1 , and t_2 leads to an increase in spiking frequency (f_{spike}), particularly notable in the case of W_2 . Conversely, f_{spike} decreases with an increase in L_1 . This pattern is attributed to the corresponding reduction in OECT capacitance with decreasing geometries. The opposite trend with f_{spike} is explained by the decrease in T_1 conductivity (and consequently drain current) with increasing f_{spike} , resulting in a lower V_{ON} , as evidenced by the amplitude of the spikes in Supplementary Fig. 12 (panel c). Voltage and current spiking amplitudes are examined in Supplementary Figs. 12 and 13, respectively. In both cases, amplitude increases with W_1 , W_2 , t_1 , and t_2 , while decreasing with L_1 , and L_2 . Supplementary Fig. 15 explores power consumption (P_{OAN}) in relation to geometries. P_{OAN} diminishes with increased W_1 , L_2 , and t_1 , whereas it decreases in other cases. Notably, reducing W_2 emerges as the most effective means to minimize power consumption. By reducing W_2 , T_2 conductivity decreases, thereby minimizing current in the R_1 - T_2 branch of the OEND. This analysis has been incorporated into the revised manuscript (pages 11-12) along with Supplementary Figs. 7, 9, 12, 13, and 15.

Supplementary Figure 7 | OEND characteristics as a function of geometrical parameters. The OEND is accessed in current mode, $V_{\text{OEND}}(I_{\text{OEND}})$. **a** Channel width transistor T_1 , W_1 , ranges from $32 \mu\text{m}$ to $63 \mu\text{m}$. **b** Channel width transistor T_2 , W_2 , ranges from $50 \mu\text{m}$ to $500 \mu\text{m}$. **c** Channel length transistor T_1 , L_1 , ranges from $25 \mu\text{m}$ to $45 \mu\text{m}$. **d** Channel length transistor T_2 , L_2 , ranges from $1 \mu\text{m}$ to $30 \mu\text{m}$. **e** Channel thickness of transistor T_1 , t_1 , ranges from 80 nm to 160 nm . **f** Channel thickness of transistor T_2 , t_2 , ranges from 80 nm to 160 nm . The dashed lines in all panels represent the load-line, aiding in showing changes of the crossing point as the parameter varies, while the arrows indicate the direction of parameter increase.

Supplementary Figure 9 | OAN spiking frequency as a function of geometrical parameters. Investigation of the spiking frequency, f_{spike} , as a function of the geometrical parameters: OMIEC channel width W_1 , W_2 , length L_1 , L_2 , and thickness t_1 , t_2 . It is observed that reducing the dimensions such as W_1 , W_2 , L_2 , t_1 , and t_2 leads to an increase in spiking frequency (f_{spike}), particularly notable in the case of W_2 . Conversely, f_{spike} decreases with an increase in L_1 . This pattern is attributed to the corresponding reduction in OECT capacitance with decreasing geometries. The opposite trend with f_{spike} is explained by the decrease in T_1 conductivity (and consequently drain current) with increasing f_{spike} , resulting in a lower V_{ON} , as evidenced by the amplitude of the spikes in Supplementary Fig. 12 (panel c).

Supplementary Figure 12 | Amplitude of OAN spiking voltage vs geometrical parameters. Investigation of the voltage spiking amplitude $A_{V_{spike}}$ as a function of OMIEC geometrical parameters: **a** width of T_1 , **b** width of T_2 , **c** length of T_1 , **d** length of T_2 , **e** thickness of T_1 , **f** thickness of T_2 .

Supplementary Figure 13 | Amplitude of OAN spiking current vs geometrical parameters. Investigation of the current spiking amplitude $A_{I_{spike}}$ as a function of OMIEC geometrical parameters: **a** width of T_1 , **b** width of T_2 , **c** length of T_1 , **d** length of T_2 , **e** thickness of T_1 , **f** thickness of T_2 .

Supplementary Figure 15 | OAN power vs geometrical parameters. Investigation of the OAN power consumption as a function of as a function of OMIEC geometrical parameters: **a** width of T_1 , **b** width of T_2 , **c** length of T_1 , **d** length of T_2 , **e** thickness of T_1 , **f** thickness of T_2 . P_{OAN} diminishes with increased W_1 , L_2 , and t_1 , whereas it decreases in other cases. Notably, reducing W_2 emerges as the most effective means to minimize power consumption. By reducing W_2 , T_2 conductivity decreases, thereby minimizing current in the R_1 - T_2 branch of the OEND.

3. *Hoe extendable is this model to other OMIEC?*

[R3Q3] Authors' response:

We appreciate the Reviewer for highlighting this crucial aspect. Indeed, this represents a key advantage of our methodology that merits further attention. Our model possesses the flexibility to encompass a range of OMIECs, achieved by integrating their material parameters into the framework outlined in Supplementary Note 1. Consequently, our numerical framework maintains self-consistency and offers a comprehensive analysis of OAN behavior and performance, tailored to the unique characteristics of the OMIEC materials utilized for transistors T_1 and T_2 .

Specifically, we found that the volumetric capacitances C_{V1} and C_{V2} of the OMIECs used as channel materials of T_1 and T_2 , respectively, are very significant material parameters

influencing f_{spike} , with the spiking frequency being maximum when C_{V1} and C_{V2} are minimized. To further explore this aspect, Supplementary Fig. 8 analyzes the OAN spiking frequency while also considering the relationship between C_{V1} , C_{V2} , and load capacitor C_L . We systematically varied both C_{V1} and C_{V2} , and the minimum C_L required for OAN spiking is calculated. Supplementary Fig. 8a reveals that the minimum C_L amounts to 10^{-7} F and is achieved when C_{V2} falls in the range 50 - 100 F cm^{-3} and C_{V1} is approximately 50 F cm^{-3} . Supplementary Fig. 8b highlights that the maximum spiking frequency, $f_{\text{spike}} = 150$ Hz, is attained when both C_{V2} and C_L are minimized. We observe that minimizing C_{V2} leads to a reduction in the capacitance of OECT T_2 , which can also be achieved by adjusting the geometrical parameters.

In Supplementary Figs. 16 and 17, the analysis of the impact of material parameters on the OAN performance is further extended to several state-of-art OMIECs suitable for depletion-mode (transistor T_1) and accumulation mode (transistor T_2) devices. To assess the impact of each material on the OAN performance, we modified one material at a time while keeping all other parameters constant. The analysis confirms that OMIECs with large volumetric capacitance and low mobility (e.g. p(gNDI-g2T) in Supplementary Fig. 16), provides OANs with reduced spiking frequency, large energy per spike and low power consumption. This is further confirmed in Supplementary Fig. 17 where the material with lowest mobility, i.e. p(gBDT-g2T), provides the lower spiking frequency and the maximum power consumption. This also suggests that the mobility of OMIECs used for T_2 is important to control the power consumption of the OAN.

According with the Reviewer's comment, we included this analysis in the revised manuscript (page 12, Figs. 4e, and 4f, Supplementary Figs. 8, 16 and 17).

Fig. 4 | OAN spiking frequency and energy. e volumetric capacitance of T_1 , **f** volumetric capacitance of T_2 .

Supplementary Figure 8 | OAN spiking frequency analysis. Investigation of the spiking frequency, f_{spike} , as a function of the volumetric capacitances C_{v1} and C_{v2} of the OECT T_1 and T_2 , respectively. **a** Minimum load capacitor, C_L , required for OAN spiking for each combination of C_{v1} and C_{v2} . **b** f_{spike} as a function of C_{v1} and C_{v2} with minimum C_L . The gray regions indicate cases of non-spiking OAN.

Supplementary Figure 16 | OAN performance with various OMIEC materials for depletion-mode OECTs (T_1). Investigation of the OAN performance considering various OMIEC materials suitable for the fabrication of depletion-mode transistor T_1 . The fabricated OAN is based on the material PEDOT:PSS for transistor T_1 and p(g2T-TT) for transistor T_2 . The star symbols indicate the materials and the corresponding parameters used in our work. The parameters of our materials are the following. PEDOT:PSS*: volumetric capacitance $C_V = 43 \text{ F cm}^{-3}$, hole mobility $\mu = 1.02 \text{ cm}^2 \text{ V}^{-1} \text{ s}^{-1}$, threshold voltage $V_{\text{TH}} = 0.477 \text{ V}$. p(g2T-TT)*: $C_V = 241 \text{ F cm}^{-3}$, $\mu = 0.13 \text{ cm}^2 \text{ V}^{-1} \text{ s}^{-1}$, and $V_{\text{TH}} = -0.23 \text{ V}$. The performance estimated with the state-of-art materials are calculated by substituting the material parameters of transistor T_1 with those of the material under investigation. The parameters of the materials investigated (triangle symbols) are taken from Refs.²⁷⁻³¹, and are the following. PEDOT:PSS: $C_V = 39 \text{ F cm}^{-3}$, $\mu = 1.9 \text{ cm}^2 \text{ V}^{-1} \text{ s}^{-1}$, $V_{\text{TH}} = 0.4 \text{ V}$. PEDOT:TOS: $C_V = 136 \text{ F cm}^{-3}$, $\mu = 0.93 \text{ cm}^2 \text{ V}^{-1} \text{ s}^{-1}$, and $V_{\text{TH}} = 0.52 \text{ V}$. p(gNDI-g2T): $C_V = 397 \text{ F cm}^{-3}$, $\mu = 1 \cdot 10^{-4} \text{ cm}^2 \text{ V}^{-1} \text{ s}^{-1}$, and $V_{\text{TH}} = 0.35 \text{ V}$.

Supplementary Figure 17 | OAN performance with various OMIEC materials for accumulation mode OECTs (T_2). Investigation of the OAN performance considering various OMIEC materials suitable for the fabrication of accumulation-mode transistor T_2 . The fabricated OAN is based on the material PEDOT:PSS for transistor T_1 and p(g2T-TT) for transistor T_2 . The star symbols indicate the materials and the corresponding parameters used in our work. The parameters of our materials are the following. PEDOT:PSS*: volumetric capacitance $C_V = 43 \text{ F cm}^{-3}$, hole mobility $\mu = 1.02 \text{ cm}^2 \text{ V}^{-1} \text{ s}^{-1}$, threshold voltage $V_{\text{TH}} = 0.477 \text{ V}$. p(g2T-TT)*: $C_V = 241 \text{ F cm}^{-3}$, $\mu = 0.13 \text{ cm}^2 \text{ V}^{-1} \text{ s}^{-1}$, and $V_{\text{TH}} = -0.23 \text{ V}$. The performance estimated with the state-of-art materials are calculated by substituting the material parameters of transistor T_2 with those of the material under investigation. The parameters of the materials investigated (triangle symbols) are taken from Refs.²⁷⁻³¹, and are the following. p(g2T-TT): $C_V = 241 \text{ F cm}^{-3}$, $\mu = 0.94 \text{ cm}^2 \text{ V}^{-1} \text{ s}^{-1}$, and $V_{\text{TH}} = -0.2 \text{ V}$. p(g2T-T): $C_V = 220 \text{ F cm}^{-3}$, $\mu = 0.28 \text{ cm}^2 \text{ V}^{-1} \text{ s}^{-1}$, and $V_{\text{TH}} = 0 \text{ V}$. p(gBDT-g2T): $C_V = 77 \text{ F cm}^{-3}$, $\mu = 0.018 \text{ cm}^2 \text{ V}^{-1} \text{ s}^{-1}$, and $V_{\text{TH}} = -0.55 \text{ V}$.

4. In the bio-interfacing testing, what is the effect of adding toxin compounds to the electrolyte medium on device performance? Testing results from a control group without cells should be provided.

[R3Q4] Authors' response: We evaluated the impact of the toxin compounds by measuring the electrical characteristics of OECT T_1 operating in PBS and in PBS with the same toxic compound used in the biointerfacing testing. Supplementary Fig. 29a shows that the electrical

characteristics measured in PBS (color dashed lines) overlap with those measured in PBS with the toxic compound. In Supplementary Fig. 29a we calculated

$$\text{Err.}\% = 100 \times [I_D(\text{PBS}) - I_D(\text{H}_2\text{O}_2)]/I_D(\text{PBS}),$$

where $I_D(\text{PBS})$ is the drain current measured with PBS only, and $I_D(\text{H}_2\text{O}_2)$ is the drain current measured in PBS with toxic compound. The maximum $\text{Err.}\% < 1.5\%$ confirming that adding toxin compounds to the electrolyte medium does not affect the device performance.

We included this analysis in the revised manuscript (page 22 and Supplementary Fig. 29).

Supplementary Figure 29 | Impact of the toxic compound on the OECT. **a** Measured output characteristics (I_D - V_D) at various gate voltages V_G in PBS (colour lines) and in PBS with addition of 10^{-3} M H_2O_2 (dotted black lines). **b** $\text{Err.}\% = 100 \times [I_D(\text{PBS}) - I_D(\text{H}_2\text{O}_2)]/I_D(\text{PBS})$, where $I_D(\text{PBS})$ is the drain current measured using only PBS as electrolyte and $I_D(\text{H}_2\text{O}_2)$ is the drain current measured using PBS with addition of 1×10^{-3} M H_2O_2 . The measurements show that the toxic compound has a negligible impact on the device.

5. A diagram of detailed device fabrication process and the details of bio-interfacing testing should be provided to enhance clarity for readers.

[R3Q5] Authors' response: According with the Reviewer's suggestion we added a diagram of detailed device fabrication process (Supplementary Fig. 32) and the details of bio-interfacing testing (Supplementary Fig. 27).

Supplementary Figure 32 | Diagram of device fabrication process. **a** Photograph of transistor channel with dimensions $W = 50 \mu\text{m}$, and $L = 20 \mu\text{m}$. **b** Standard microscope glass slides ($75\text{mm} \times 25\text{mm}$) were cleaned in a sonicated bath, first in soap solution (Micro-90 (Sigma-Aldrich)) and then in a 1:1 (vol/vol) solvent mixture of acetone and isopropanol. Source and drain electrodes were made with photolithographically patterned gold (with positive Microposit S1813 photoresist (DOW)) on the cleaned glass slides. A chromium layer was used to improve the adhesion of gold. Two layers of parylene C (SCS Coatings) were deposited. Soap (Micro-90 soap solution, 1% vol/vol in deionized water) was used for separation between the parylene C layers to enable the peel-off of the upper parylene C layer. The lower parylene C layer insulates the gold electrodes. Silane A-174 (γ -methacryloxypropyl trimethoxysilane) from Sigma-Aldrich was added to the lower parylene C layer to enhance adhesion. The channel dimensions of the transistors were defined in the second photolithography step through the positive photoresist AZ 9260 MicroChemicals (Cipec Spécialités). Reactive ion etching (O₂/CF₄ plasma, 160W for 16min with O₂ flow rate of 50s.c.c.m. and CHF₃ flow rate of 5s.c.c.m.) was used to define the transistor channels throughout the photoresist mask. Channel of transistor T₁ is made with the organic mixed ionic–electronic conductor polymer PEDOT:PSS (Clevios PH 1000) mixed with 5.0wt% ethylene glycol, 0.1wt% dodecyl benzene sulfonic acid and 1.0wt% (3-glycidylxypropyl)trimethoxysilane. Spin coating was used to produce a film in two steps at 1,500rpm and 650rpm for 1min and annealed at 120°C for 1min in between. Channel of transistor T₂ is made with the semiconducting polymer p(g2T-TT), dissolved in chloroform (3mgml^{-1}) inside a N₂-filled glovebox and spin coated in ambient conditions at 1,000rpm for 1min resulting in a thickness of 40nm. The devices were baked at 60°C for 1min. The sacrificial upper parylene C layer was peeled off to confine the polymer to the inside of the channel regions. The devices were subsequently baked at 140°C for 1hour. Excess soap was rinsed off with deionized water. **c** Photograph of the OAN device.

Supplementary Figure 27 | Schematic of the biohybrid OAN. **a** Circuit diagram of the biohybrid OAN. **b** Photograph of the cellular membrane integrated with the OAN. **c** Simplified cross-section of T1 with the cellular membrane in the trans-well filter. The gate is inserted in the electrolyte of the apical compartment and the channel is in contact with the electrolyte of the basal compartment. Cell medium is used as electrolyte. **d** Schematic of the epithelial cell layer acting as a physical barrier against ion movement between the basal and apical domains. The cells are interconnected by tight junctions (TJs), forming a barrier that restricts ion flow across the membrane. Ion movement occurs through two potential pathways: the transcellular route (through the cells) and the paracellular route (between the cells). Toxins like hydrogen peroxide (H_2O_2) can disrupt the TJs, impairing the paracellular pathway for ion transport. The equivalent circuit of the bio-membrane is represented by a parallel $R_M C_M$ circuit, with R_M denoting the transmembrane resistance and C_M representing the membrane capacitance. Toxins such as H_2O_2 can compromise the integrity of the TJs, leading to increased transmembrane ion permeability and a reduction in both R_M and C_M . In a biohybrid neuron initially biased below the firing threshold, the introduction of H_2O_2 disrupts the TJs of the bio-membrane, consequently initiating firing.

We are grateful for the Reviewer's insightful comments and suggestions, which helped us to significantly improve the clarity, precision, and depth of the research. By considering and incorporating the Reviewer's feedback and suggestions, we have expanded the scope of our investigations and analyses, thus ensuring that our work is more comprehensible and relevant to a broader scientific audience.

REVIEWERS' COMMENTS

Reviewer #1 (Remarks to the Author):

Authors were able to respond to all concerns - congratulations!

Reviewer #2 (Remarks to the Author):

The authors provided an in depth review answering all the point raised during the first review.

The presentation of the impact has been marginally strenghtened in the introduction and would constitute a minor revision to improve further the paper.

Technical comment:

I would have included the table 1-SI into the main text. The table should be also better discussed or more exhaustive, in particular, what means columns 5, 6 and 7: there is lots of Si transistors devices that are used for ion sensing and for biological interfacing, why all Si technology are considered "no"?. This element needs to be clarifoied or corrected since it is a pivotaql argument for the paper

Reviewer #2 (Remarks on code availability):

n.a.

Response to the Reviewers

We thank the Reviewers for their comments.

Reviewer #1 (Remarks to the Author):

Authors were able to respond to all concerns - congratulations!

Authors' response: We thank the Reviewer.

Reviewer #2 (Remarks to the Author):

The authors provided an in depth review answering all the point raised during the first review. The presentation of the impact has been marginally strenghtened in the introduction and would constitute a minor revision to improve further the paper. Technical comment: I would have included the table 1-SI into the main text. The table should be also better discussed or more exhaustive, in particular, what means columns 5, 6 and 7: there is lots of Si transistors devices that are used for ion sensing and for biological interfacing, why all Si technology are considered "no"? This element needs to be clarified or corrected since it is a pivotal argument for the paper

[R2Q1] Authors' response: We are glad that the Reviewer appreciated our work. According to the Reviewer's comment, we replaced columns 5, 6 and 7 of Supporting Table 1 with a new column with the following title: "Neuron in-liquid ion-sensing and bio-interfacing". This clarifies that we considered artificial neurons capable to operate directly in liquid environment and their spiking properties depend on the ion species and their concentration as well as on the biological status of cells. Moreover, we extended the caption of Supplementary Table 1 providing a description of the various entries of the table (page 4 Supporting Information).

We agree with the Reviewer that ion-sensing transistors are widely used for ion sensing. However, Supplementary Table 1 focuses on the properties in the level of artificial neurons. To the best of our knowledge, biochemical-dependent spiking behavior of silicon artificial neuron is still underexplored, and this is what the table reflects.

According to the Reviewer comment, for the sake of clarity, we included a subset of Supplementary Table 1 in the main text (Table 2, pages 25 of the manuscript). Table 2 focuses on the artificial spiking neurons providing biophysical realism capabilities.

Thank you once again for your valuable feedback and consideration of our manuscript.

Reviewer #3 (Remarks to the Author):

No comments

Authors' response: We thank the Reviewer.